# Spatial transcriptomics using combinatorial fluorescence spectral and lifetime encoding, imaging and analysis

Tam Vu[1,2,12], Alexander Vallmitjana [1,3,12], Joshua Gu[2,4,12], Kieu La[1], Qi Xu [1], Jesus Flores[2,5], Jan Zimak[6], Jessica Shiu[7], Linzi Hosohama[8], Jie Wu[4,9], Christopher Douglas[10], Marian L. Waterman [8,9], Anand Ganesan[4,7,9], Per Niklas Hedde [3,6,11], Enrico Gratton [1,3,11✉] & Weian Zhao [1,2,4,6,9✉]

Multiplexed mRNA profiling in the spatial context provides new information enabling basic research and clinical applications. Unfortunately, existing spatial transcriptomics methods are limited due to either low multiplexing or complexity. Here, we introduce a spatialomics technology, termed Multi Omic Single-scan Assay with Integrated Combinatorial Analysis (MOSAICA), that integrates in situ labeling of mRNA and protein markers in cells or tissues with combinatorial fluorescence spectral and lifetime encoded probes, spectral and time-resolved fluorescence imaging, and machine learning-based decoding. We demonstrate MOSAICA's multiplexing scalability in detecting 10-plex targets in fixed colorectal cancer cells using combinatorial labeling of five fluorophores with facile error-detection and removal of autofluorescence. MOSAICA's analysis is strongly correlated with sequencing data (Pearson's r = 0.96) and was further benchmarked using RNAscope[TM] and LGC Stellaris[TM]. We further apply MOSAICA for multiplexed analysis of clinical melanoma Formalin-Fixed Paraffin-Embedded (FFPE) tissues. We finally demonstrate simultaneous co-detection of protein and mRNA in cancer cells.

[1] Department of Biomedical Engineering, University of California, Irvine, Irvine, CA 92697, USA. [2] Sue and Bill Gross Stem Cell Research Center, University of California, Irvine, Irvine, CA 92697, USA. [3] Laboratory for Fluorescence Dynamics, University of California, Irvine, Irvine, CA 92697, USA. [4] Department of Biological Chemistry, University of California, Irvine, Irvine, CA 92697, USA. [5] CIRM Stem Cell Research Biotechnology Training Program at California State University, Long Beach, Long Beach, CA 90840, USA. [6] Department of Pharmaceutical Sciences, University of California, Irvine, Irvine, CA 92697, USA. [7] Department of Dermatology, University of California, Irvine, Irvine, CA 92697, USA. [8] Department of Microbiology and Molecular Genetics, University of California, Irvine, Irvine, CA 92697, USA. [9] Chao Family Comprehensive Cancer Center, University of California, Irvine, Irvine, CA 92697, USA. [10] Department of Pathology & Laboratory Medicine, University of California, Irvine, Irvine, CA 92617, USA. [11] Beckman Laser Institute & Medical Clinic, University of California, Irvine, Irvine, CA 92697, USA. [12]These authors contributed equally: Tam Vu, Alexander Vallmitjana, Joshua Gu. ✉email: egratton22@gmail.com; weianz@uci.edu

Cell fate and cell-cell, cell-niche interactions are tightly regulated in space at both genetic and tissue and system level to mediate organ development, tissue homeostasis and repair, and disease appearance and progression. Therefore, spatial transcriptomics that profile gene expression landscape at the single-cell level in tissues in a 3D spatial context as shown in this work represents a frontier in biological research and precision medicine[1–8]. For instance, spatial transcriptomics techniques can (a) help realize the vision of the human cell atlas in generating "high-resolution and comprehensive, three-dimensional reference maps of all human cells in the body", (b) determine molecular mechanisms that govern cell fate, state, lineage and cell cooperation in tissue formation in developmental biology and regenerative medicine, (c) investigate the biological changes associated with different diseases in a spatial-dynamic fashion and to uncover disease molecular mechanisms and discover disease biomarkers, and (d) characterize the complexities of tissue biopsy (e.g., tumor) in clinical pathology to inform personalized disease diagnosis and therapeutic intervention in the era of precision medicine. Spatial transcriptomics tools need to be able to assess multiple transcripts within the same cell and sample in a highly multiplexed fashion due to the heterogeneous gene expression and many different cell identities/states exist in a particular tissue. Furthermore, patient derived materials are often available in limited quantity and generating many sections to test for different markers separately is tedious and non-feasible.

A major bottleneck in spatial transcriptomics is the lack of tools that can be both easy-of-use and highly multiplexing[7–13]. Conventional tools for in situ analysis including fluorescence in situ hybridization (FISH) (e.g., LGC Stellaris™) can only detect 3–4 targets at a time because of the limited number of spectral channels in fluorescence microscopes[12–14]. Conventional methods for in situ profiling of transcripts are further confounded by the autofluorescent moieties in tissue preparations including clinical biopsies. Recent single-cell RNA sequencing methods provide information on the presence and identity of transcripts in single cells but lack the critical spatial context needed to understand complex heterogeneous tissue[15–17]. Imaging- and FISH-based spatial transcriptomic methods that employ sequential labeling, stripping, and imaging (e.g., seqFISH, MERFISH) or branched amplification (e.g., RNAscope™, SABER) are often complicated, error-prone, time-consuming, laborious and/or costly to scale up[18–22]. Furthermore, repeated processing of the same sample can in some cases affect tissue structural integrity and target molecules and may not always be amenable for clinical applications such as profiling patient biopsies. Spatial transcriptomics using in situ sequencing (e.g., ISS, FISSEQ, starMAP and ExSeq) or in situ barcoding coupled with ex situ sequencing (e.g., GeoMx, slide-seq, and DBiT-seq) can drastically improve multiplexing but suffer from reduced spatial resolution and detection efficiency especially for low-abundance targets[22–25].

In this work, we report a fluorescence imaging-based spatial-omics technology termed MOSAICA (Multi Omic Single-scan Assay with Integrated Combinatorial Analysis) that enables direct, highly multiplexed biomarker profiling in the 3D spatial context in a single round of staining and imaging. MOSAICA employs in situ staining with combinatorial fluorescence spectral and lifetime encoded probes, spectral- and time-resolved fluorescence imaging, and AI-based target decoding pipeline (Fig. 1). Fluorescence lifetime is a measure of the time a fluorophore spends in the excited state before returning to the ground state and is an inherent characteristic of the fluorophore and its surrounding environment[26,27]. By utilizing both time and spectral domains for labeling and imaging, we were able to discriminate a repertoire of 10 different fluorescent signatures against auto-fluorescent moieties and nonspecific binding events within the

same sample in this study and expect to scale up to at least 60-plex in the future to enable increased multiplexing capabilities with standard optical systems.

In this study, we describe the MOSAICA pipeline, including automated probe design algorithm, probe hybridization optimization, and validation, combinatorial spectral and lifetime labeling and analysis for target encoding and decoding. Particularly, we developed an automated machine learning-powered spectral and lifetime phasor segmentation software that has been developed to spatially reveal and visualize the presence, identity, expression level, location, distribution, and heterogeneity of each target mRNA in the 3D context. We showcased MOSAICA in analyzing a 10-plex gene expression panel in colorectal SW480 cells based on combinatorial spectral and lifetime barcoding of only five generic commercial fluorophores. Using this model, we illustrated the multiplexing scalability and MOSAICA's ability to correct for stochastic nonbinding artifacts present within the sample. We further demonstrated MOSAICA's utility in improved multiplexing, error-detection, and autofluorescence removal in highly scattering and autofluorescent clinical melanoma FFPE tissues, demonstrating its potential use in tissue for cancer diagnosis and prognosis. To further reveal the potential of MOSAICA, we demonstrated its multiomics capability with simultaneous co-detection of protein and mRNA in colorectal SW480 cells. MOSAICA is rapid, cost-effective, and easy-to-use and can fill a critical gap between conventional FISH and sequential- and sequencing-based techniques for targeted and multiplexed spatial transcriptomics.

## Results

**MOSAICA workflow.** In a typical MOSAICA workflow (Fig. 1), primary oligonucleotide probes designed to specifically bind to mRNA targets with a complementary target region (25–30 base long) are incubated with fixed cell or tissue samples (Fig. 1a, b). These primary probes also contain an adjacent adaptor region consisting of two readout sequences for modular secondary probe binding. In this study, double-ended secondary probes with fluorophores on each end are hybridized to the readout region on the primary probes (Fig. 1c). Through combinatorial labeling, each target is encoded with a dye with a distinct spectrum and lifetime signature. The labeled samples are then imaged using a custom built or commercial microscope (e.g., the Leica SP8 Falcon used in this study) equipped with spectral and lifetime imaging capabilities (Fig. 1d). Both spectral and fluorescence lifetime data will be captured, and then analyzed using phasor plots (Fig. 1e). Our automated machine learning algorithm and a codebook finally reveal the locations, identities, counts, and distributions of the present mRNA targets in a 3D context (Fig. 1f).

**Probe design pipeline.** To rapidly design oligonucleotide probes for the transcript of each gene, we modified the python platform, OligoMiner[28], a validated pipeline for rapid design of oligonucleotide FISH probes. Briefly, as shown in Supplementary Fig. 1a, using the mRNA or coding sequence file of the target gene, the blockParse.py script will screen the input sequence and output a file with candidate probes while allowing us to maintain consistent and customized length, GC, melting temperature, spacing, and prohibited sequences. Using Bowtie2, the candidate probes are rapidly aligned to the genome to provide specificity information that is used by the outputClean.py script to generate a file of unique candidates only. The primary probes comprise complementary sequence of typically 27–30 nucleotides and are designed mostly within the coding sequence region, which has fewer variation than the untranslated region[20]. We wrote a script, seqAnalyzer.py, to automate the alignment of primary probes to

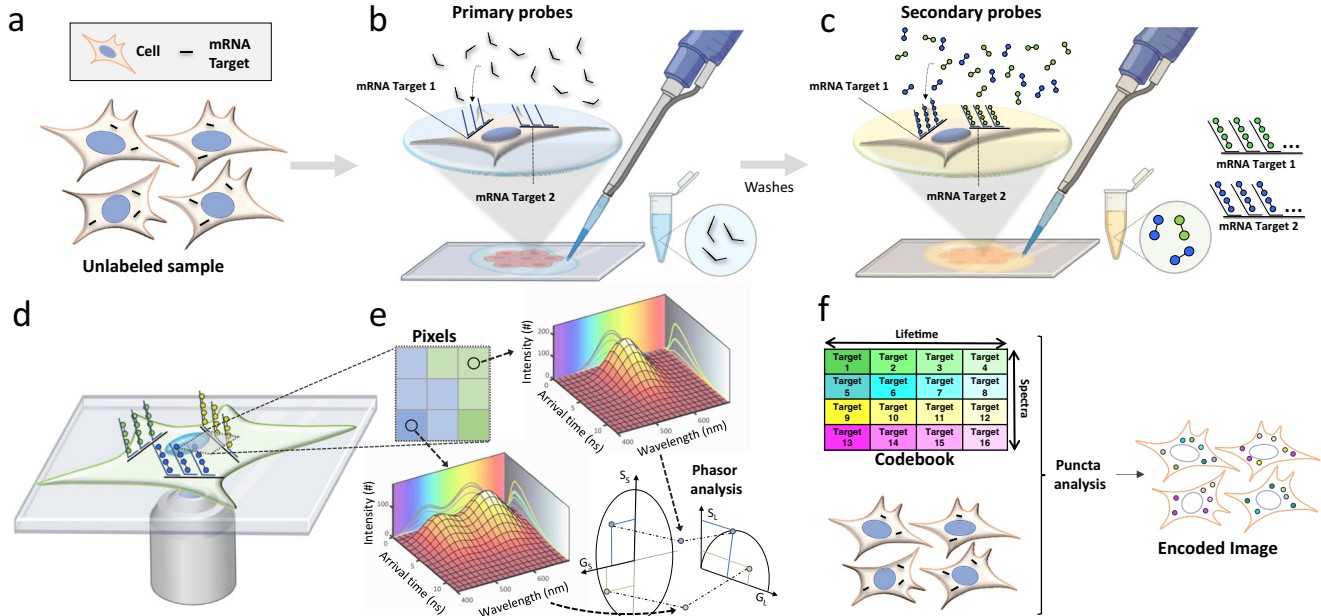

**Fig. 1 Schematic of the MOSAICA approach for labeling and analysis of spectral and time-resolved components. a** Sample(s) can be fixed cells or tissues. RNA transcripts from genes of interest are targeted for detection. Protein targets can be stained too in mRNA and protein codetection. **b** Primary labeling probes are designed to include two functional regions: a target region which is complementary and can bind to the mRNA target and an adjacent readout region which can subsequently bind to fluorescently labeled oligonucleotides. **c** Secondary fluorescent probes are added to bind to the primary probes to form different combinations (combinatorial labeling) through a "readout" domain. **d** Labeled targets are measured under a fluorescent microscope to interrogate the spectral and lifetime characteristics of the labeled moieties. **e** Phasor analysis is used to identify which fluorophore labels are present in each pixel and puncta. **f** Labeled targets eliciting the encoded intensity-based and time-based signature are decoded to reveal the locations, identities, counts, and distributions of the present mRNA targets in a multiplexed fashion.

sequencing data (Supplementary Fig. 1b) so that probes that aligned to regions of lower read counts would be discarded. Furthermore, primary probe "readout" domains and secondary probes (typically 15–20 nucleotides long) are designed to be orthogonal to each other to avoid off-target binding. Libraries and databases of over 200,000 orthogonal sequences are available online and we have simply used those that have been previously validated[29]. Fluorophores exhibiting distinct spectrum (typically with excitation/emission spectra in the 400–700 nm range) and lifetimes (typically in the 0.3–10 ns range) can be conjugated to oligos which were obtained through commercial vendors (see Methods).

**Probe labeling validation and optimization**. We first investigated the specificity of our labeling condition using a simple cell mixture model comprising wild-type HEK293T-X cells and HEK293T-X cells engineered with *mNeonGreen* (Supplementary Fig. 2a) by detecting *mNeonGreen* mRNA as the gene expression target. Since only fluorescent *mNeonGreen* positive cells can express the corresponding mRNA transcripts, this cell mixture model provides a straightforward tool to assess the specificity and nonspecific binding. Using a Nikon epifluorescence microscope to image the samples following staining with primary and secondary probes (all probe sequences used in this study are provided in Supplementary Data 1), we detected on average 43.5 puncta per *mNeonGreen* positive cell ($n = 76$ cells) and 0.25 puncta per wild-type cell ($n = 164$) (Supplementary Fig. 2b, c), indicating minimal nonspecific binding with our probe labeling strategy. To further validate the baseline level of nonspecific binding, we included a negative control with the primary probe designed toward dopachrome tautomerase, a gene in the mouse genome that is not expressed in our HEK293T-X model system, along with a condition with secondary probes only. Similarly, an

average of 43.5 puncta per cell was detected for the *mNeonGreen* cells while the wild type and negative controls a mean of 2.5 puncta per cell was detected with a lower signal-to-noise. We next optimized labeling efficiency by testing the number of primary probes and incubation times of primary probes and secondary probes (Supplementary Fig. 3). We determined our optimal condition to comprise a minimal of at least 12 primary probes for each target mRNA (in practice, we always maximize the number of primary probes per mRNA depending on the size of mRNA). Indeed, 40 primary probes per channel per mRNA were subsequently used in this study, with incubation time of 16 h for primary probe hybridization and 1 h for secondary probe hybridization, respectively, which were used in subsequent experiments.

**Imaging and phasor analysis**. Lifetime imaging is a tool that measures the spatial distribution of probes with different fluorescence lifetime. Samples are stimulated with modulated or pulsed lasers at a particular frequency, typically around the 40–80 MHz, which allows the fluorescence to decay within the stimulated period, typically in the ns range. After acquiring for sufficient time, i.e., after enough laser pulses or periods, one can construct a histogram of photon arrival times at each pixel. The shape of this histogram has a rapid rise, followed by a faster or slower decay which is characteristic of the fluorescent molecule(s) present in the pixel. To model this decay data, an exponential decay model can be fitted or alternatively one can make use of the fit-free phasor approach[30,31]. We used this second approach because it requires no a priori knowledge of an underlying model (i.e. number of fluorescent species at the pixel) and it is computationally inexpensive in virtue of the Fast Fourier Transform algorithm. The phasor transform extracts two values from the decay curve that characterize the shape (and importantly not the

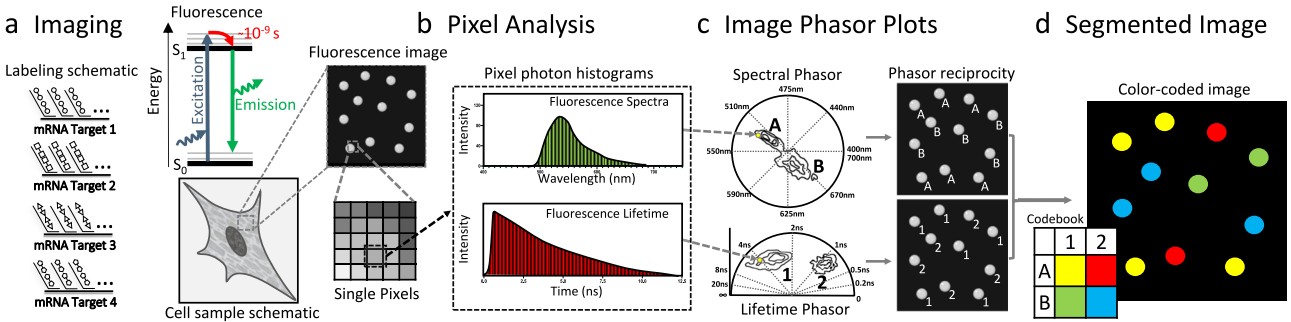

**Fig. 2 Image and phasor analysis with spectrum and lifetime analysis in MOSAICA. a** As an example, four different probes are used to target the transcripts of four different genes. The fluorescence is collected using the spectral and Fluorescence Lifetime Imaging and Microscopy (FLIM) instrument to form images where each pixel carries information of the spectra and lifetime. **b** At each pixel we compute the photon distribution in the spectral and temporal dimension. The phasor transform maps these distributions in each pixel to a position on the phasor space. **c** The phasor plots reveal the presence of different populations. These populations are identified and then mapped back to the original image. **d** We color code the pixels based on the combination of the two properties. This allows us to separate by lifetime probes that were emitting with similar spectra and vice-versa, separate by spectra probes that fluoresce with similar lifetimes.

size, so that the transform is independent of the amount of photons) and these two values, namely S and G, correspond to the two coordinates of the pixel on the phasor plot (see equations in Supplementary Note 1). The values are obtained by an integral of the product of the decay of the two trigonometric functions, sine and cosine, fit in the stimulation period, and they correspond to the first-order terms of the Fourier Series decomposition of the decay curve.

Similarly, if one uses a spectral detector, i.e., a separate detector for different spectral bands, then for each pixel, one can obtain another histogram, in this case with the number of photons arriving in each channel, i.e., at each wavelength. This curve can also be transformed to an analogous spectral phasor space to map the recorded spectra at each pixel onto the 2D spectral phasor space[32,33]. Combining the lifetime measurement with a spectral detector, one effectively has a 5-dimensional space in which to characterize each pixel. On top of the spatio-temporal coordinates (x,y,z,t), each pixel now carries information in five additional coordinates: its intensity value (however many photons arrived at that pixel), the two phasor coordinates for the lifetime phasor transform, and the two phasor coordinates for the spectral phasor transform[34]. A typical image, on the order of $10^6$ pixels, obtained with this method provides $10^6$ points in this 5D space[34]. If the sample presents different populations of fluorescent molecules at different locations, the pixel phasor data at these different locations map to different positions in this phasor space and a clustering technique can be used to resolve each population[35].

There is a direct analogy between the phasor transform in spectral and lifetime fluorescence microscopy (Fig. 2). As an example in this figure, we use a hypothetical experiment where transcripts from 4 different target genes are targeted with 4 fluorescent species. Of the 4 species, we construct the example so that two fluorescent species emit in one color and the other two in another color. At the same time, within each color, one has a short lifetime and the other has a long lifetime. This hypothetical sample is excited, and the individual photons are detected at each pixel (Fig. 2a). In each pixel, we accumulate enough photons to build a spectral histogram and a lifetime histogram (Fig. 2b). These curves are phasor-transformed to reveal two distinct populations in the phasor space, corresponding to the two colors and the two lifetimes. By means of our previously published automatic clustering using machine learning[35], we identify these populations and return to the image space to label each pixel depending on the group it belongs to in the phasor space (Fig. 2c). By combining the spectral and lifetime information, we

have automatically segmented the image into regions, i.e., identified the pixels that belong to the different species (Fig. 2d). Again, note that in this example in Fig. 2, we have chosen the probes to be the most convoluted case possible; one couple shares a similar spectrum and the other couple shares another spectrum. At the same time, one of the members of either couple share a similar lifetime and the other two members of either couple share another lifetime. This is the reason why even if there are four distinct fluorescent probes, only two spectral populations are detected both in the spectral and lifetime phasor space, and the combinations of these two populations yield to the four distinct groups. The four probes cannot be resolved unless both the lifetime and spectral information are accessed.

**Combinatorial target spectral and lifetime encoding and decoding.** In the previous section, we showed how by combining the time dimension with the spectral dimension, we can increase the number of possibilities and therefore enhance the multiplexing capabilities squaring the number of targets that can be resolved. To further increase multiplexing and improve detection efficiency, we employ combinatorial labeling, a method in which targets are labeled with two or more unique fluorophores, to greatly increase the base number of targets we can label with a given number of fluorophores/probes. To illustrate this concept, here we demonstrate a minimal exemplary working example of combinatorial labeling where two probes are used to label three targets. In this situation, each probe labels one target and the third target is labeled with both probes simultaneously. Figure 3 shows a real case with such configuration, both for spectra and for lifetime. The cartoon represents the case of using two probes with distinct spectra. When imaging this sample, we can use two spectral channels, Fig. 3b, c, where some targets appear in only one channel, other targets appear in only the other channel and the target that is labeled with both probes appears in both channels. All targets are then detected and color-coded depending on their presence in one channel, the other or the two simultaneously (Fig. 3d) and the overall counts of each combination in the field of view can be provided (Fig. 3e).

Similarly, we show a case in which the targets are now labeled with two probes that have similar spectra but different lifetimes (Fig. 3f). In this case, we also introduce the use of the phasor approach to reveal the three expected populations, the pixels that contain both probes appear in the midpoint between the phasor positions of the pixels that contain only one of the probes.

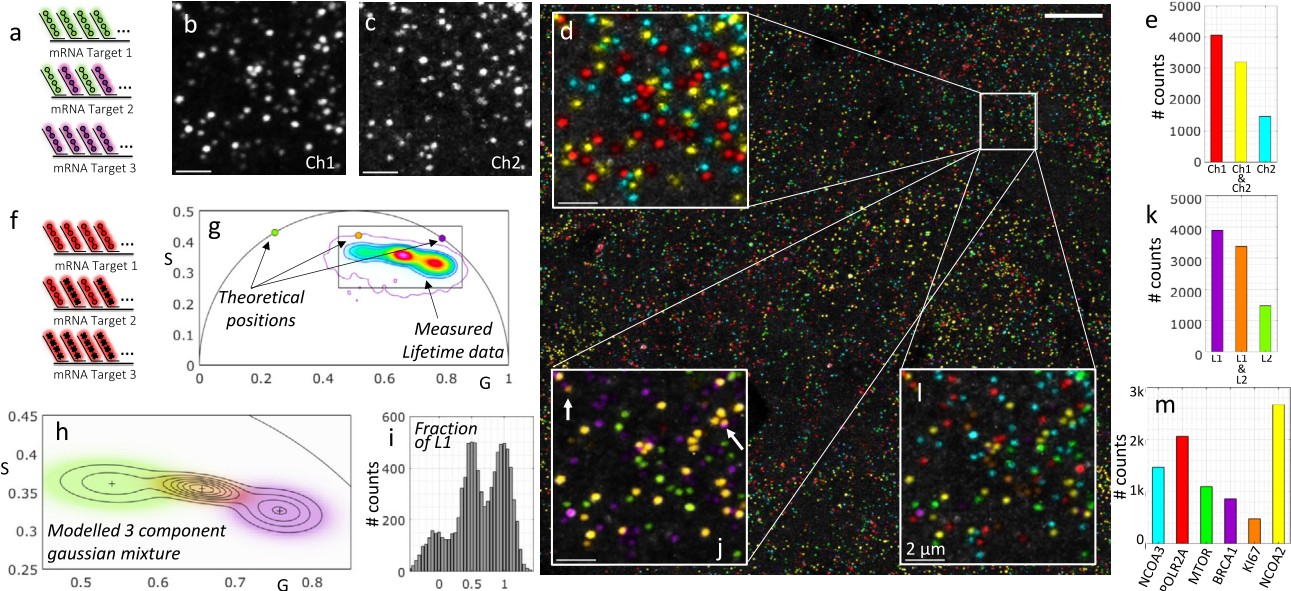

**Fig. 3 Working example of combinatorial labelling of three mRNA targets with two probes. a** Transcripts of three different target genes are tagged using two probes with different spectra. Targets 1 and 3 are tagged each with one probe, Target 2 is tagged with both simultaneously. **b**, **c** The fluorescence is collected in the two expected spectral channels for the known emission of the two probes (representative small regions of a whole 3D field of view). **d** The maximum projection of the two channels is shown and pseudo-colored depending on the presence in the respective channels (as an inset within the whole field of view. **e** The actual counts of each target within the whole field of view. **f** As a parallel example, transcripts of three different target genes are tagged using two probes with different lifetime. Targets 1 and 3 are tagged each with one probe, Target 2 is tagged with both simultaneously. **g** The phasor plot presents three populations, corresponding to the pixels with the three combinations; the two components by themselves plus the linear combination falling in the middle. **h**) Machine learning clustering technique is used to identify the groups (Gaussian mixture model). **i**) The multicomponent method is used to extract the fraction of one of the components in each detected puncta. **j** The same inset is shown with the pseudocoloring now depending on the lifetime clustering. **k** The counts for each lifetime cluster in the whole field of view. **l** The combination of the information in both the spectral and the lifetime dimension yields a final 6-plex. **m** The overall counts for the 6-plex detection of transcripts including POLR2A (Alexa647 & ATTO565), MTOR (ATTO647 & ATTO565), KI67 (Alexa647 & ATTO647), BRCA1 (Alexa647), NCOA2 (ATTO647), NCOA3 (ATTO565) with the appropriate expressed genes that correspond to each combination. Experiments were conducted with cultures of mNeon green cells. Scale bar 10 µm in large image and 2 µm in insets. Source data are provided as a Source Data file.

Figure 3g shows the phasor distribution obtained from the same field of view as in the spectral example, in which we also show the theoretical locations of the probes (corresponding to Alexa647 and ATTO647 with respective lifetimes of 1 ns and 3.5 ns). As is expected in real experimental conditions, there are additional fluorescent components in the sample. We broadly refer to the bulk of these additional components as autofluorescence, which pulls the data away from the expected positions and converges to the mean phasor position of the autofluorescent components. We have previously shown that the Gaussian Mixture Models is the most optimal machine learning clustering algorithm to model phasor data[35], and we use this machine learning technique to infer the phasor locations of the probe combinations (Fig. 3h). We can now successfully classify each pixel of the original image into one of the clusters and obtain a probability of belonging to each, i.e., the posterior probability of the model. This allows us to color code the transcripts depending on their assignment to one of the three clusters (Fig. 3j) and obtain the counts of the three-lifetime components (Fig. 3k). Additionally, we apply our lifetime multicomponent analysis technique[36] in which for each detected puncta, we estimate the presence of one of the lifetime components, in this case lifetime1 (Alexa647, purple in the figure), to obtain the expected result; that there are clearly three populations with respective fractions centered around [0, ½, and 1] (Fig. 3i).

In the general case, we combine the lifetime and spectral dimensions, and we perform the clustering of the data in a 4D spectral/lifetime phasor space. The clustering technique has the power to not only identify which puncta belong to each cluster but also to assign a probability of belonging to that cluster, which can be used to quantify the certainty of the labeling. For example, in the inset in Fig. 3j, we show two cases of puncta that have relatively low confidence in the cluster assignment; they are depicted with blended colors because they fall in the regions of the phasor space where the two clusters are merging.

In this combinatorial example in Fig. 3, the three clusters in the lifetime domain multiplexed with the channel-based in the spectral domain yield a 6-plex image using only 3 probes (Fig. 3l, m). The specific transcripts for genes targeted for this experiment with the combined probes were *POLR2A* (Alexa647 & ATTO565), *MTOR* (ATTO647 & ATTO565), *KI67* (Alexa647 & ATTO 647), *BRCA1* (Alexa647), *NCOA2* (ATTO647), *NCOA3* (ATTO565). In the general combinatorial experiment using couples of N probes the total number of possible target genes grows quadratically:

$$\binom{N}{2} = \frac{N!}{2(N-2)!} = \frac{N^2 - N}{2} \qquad (1)$$

**Simultaneous 10-plex mRNA detection in fixed colorectal cancer SW480 cells using MOSAICA.** We next applied MOSAICA to a 10-plex panel of mRNA targets in colorectal cancer SW480 cell culture samples. This cell line was chosen because its xenograft model exhibits spatial patterns of hetero-geneity in WNT signaling[37], which will allow us to study

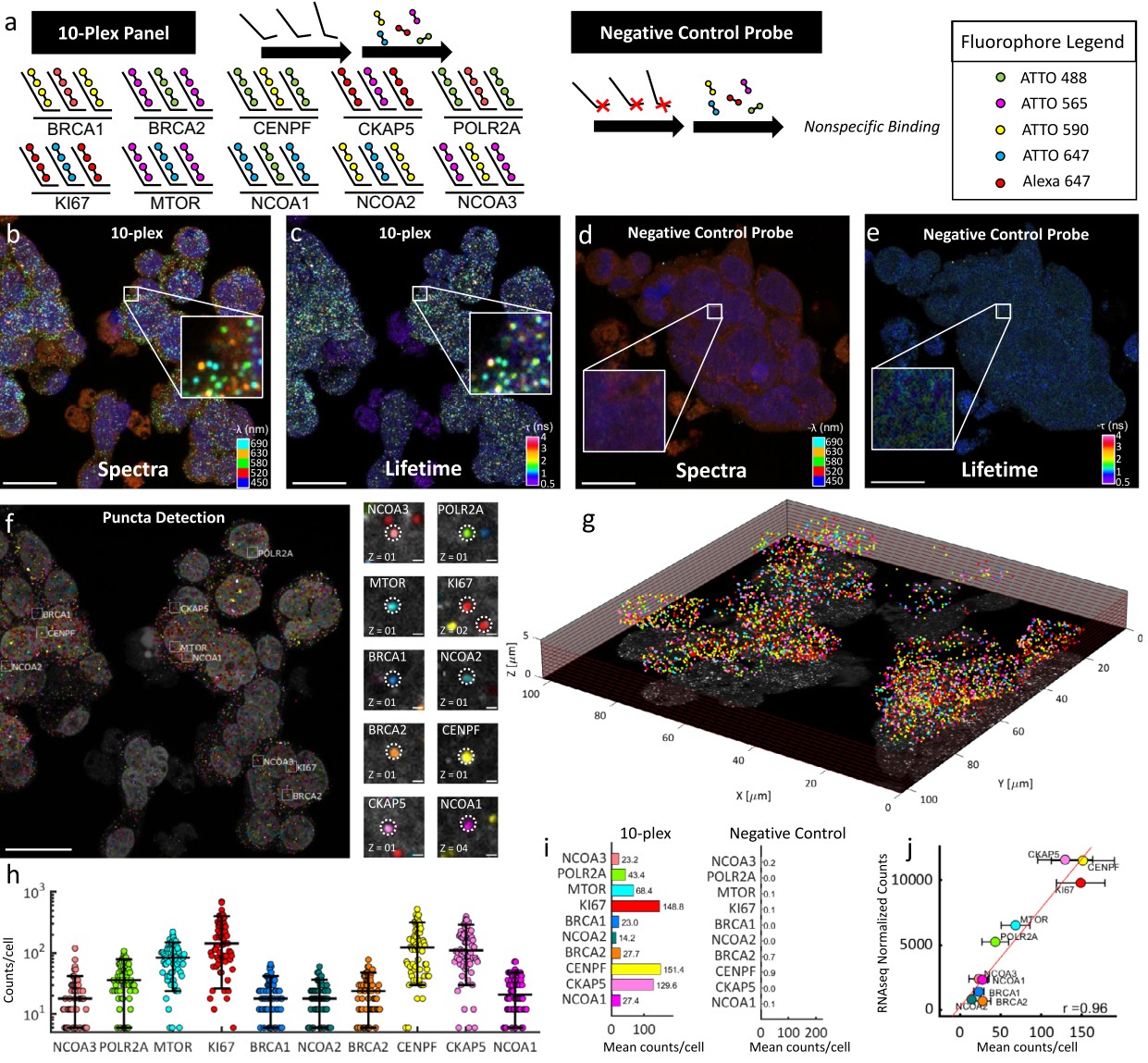

**Fig. 4 Simultaneous 10-plex detection of transcripts for genes in colorectal cancer SW480 cells in a single round of labeling and imaging. a** 10 different gene transcripts are labeled with primary probes followed by respective and complementary fluorescent secondary probes. Each transcript is labeled with a combination of 2 out of 5 fluorophores for 10 combinations. Negative control probes (mNeonGreen, DCT, TYRP1, and PAX3) targeting transcripts not present in the sample were used with their respective secondary fluorophore probes. **b** Spectral image (max-projection in z) of a field of view of the labeled 10-plex sample (5-channel pseudo coloring). **c** Lifetime image (max-projection in z) of a field of view of the labeled 10-plex sample (phasor projection on universal circle pseudo coloring). **d** Spectral image of the labeled negative control probe sample. **e** Lifetime image of the labeled negative control probe sample. **f** Final puncta detection after being processed in our analysis software showing highlighted example puncta of each target (insets, right). **g** 3D representation of the field of view for the 10-plex sample. **h** Number of puncta detected for each gene target expression in each cell for the labeled 10-plex samples (overlaid lines correspond to quantiles [10,50,90]%, n=364 cells). **i)** Mean puncta counts per cell of transcripts for each gene in the 10-plex samples (left, n=3 experimental replicates, 364 total cells profiled) and negative control probe samples (right, n=3 experimental replicates, 189 total cells profiled). **j** Correlation of detected puncta (mRNA puncta count) vs. RNA-bulk sequencing (normalized counts) is shown for each target (mean +/− standard deviation, n=3 experimental replicates), yielding a correlation (Pearson r) of 0.96. Scale bar 20 μm in large images and 1 μm in insets. Source data are provided as a Source Data file.

tumorigenesis in the spatial context and potentially identify cancer stem cell populations in colorectal cancer in future studies. Here, we selected this model as a validation platform to demonstrate the multiplexing scalability and error-detection capabilities of our approach. We began by first identifying a set of 10 genes with known expression levels from our bulk sequencing data. Using the aforementioned probe design pipeline, we designed 80 probes (two pairs of 40 probes) for the transcript of each gene: *BRCA1, BRCA2, CENPF, CKAP5, POLR2A, KI67, MTOR, NCOA1, NCOA2,* and *NCOA3*. These genes were chosen

due to their housekeeping status or involvement in tumorigenesis in colorectal cancer. By encoding the transcript of each gene with a distinct combination of two fluorophores, we generated a codebook of 10 labelling combinations from only five fluorophores following Eq. 1: $\binom{5}{2} = 10$ (Fig. 4a) (see Supplementary Table 1 and Supplementary Table 2 for the fluorophores and probes, respectively, used for each target). To assess the baseline nonspecific binding events of our assay, we included a negative probe control sample, which was labelled with primary probes

not targeting any specific sequence in the human genome or transcriptome but still containing readout regions for secondary fluorescent probes hybridization (Fig. 4a, right). Matching numbers and concentrations of primary and secondary probes that were used in the 10-plex panel were used in this sample.

An example field of view is shown in Fig. 4; first the spectral image overlay (five fluorescent channels including DAPI) of the labeled 10-plex SW480 sample (Fig. 4b) and additionally, in the same measurement, the orthogonal lifetime information attained by interrogating each pixel for their lifetime components (Fig. 4c). These pixels were phasor-transformed and pseudo-colored based on their projected phasor coordinates on the universal circle. In doing so, both dimensions of data can now be simultaneously accessed to determine which cluster of pixels meet the appropriate and stringent criteria for puncta classification. Similarly, the composite spectral and lifetime images of the corresponding negative control probe sample are shown (Fig. 4d, e). Figure 4f depicts the now detected pseudo-colored clusters which were successfully classified as one of the RNA markers. A representative inset image for each marker and its targeted detection is provided on the right. Because these are image stacks, the segmentation provides a 3D spatial distribution of the field of view, which can be rendered to visualize the spatial analysis in a 3D context (Fig. 4g).

MOSAICA employs an error-detection strategy that gates for specific and pre-encoded fluorophore combinations and rejects any fluorescent signatures which do not meet these criteria. For instance, of the total detected puncta ($n = 65,562$), we observed a considerable fraction of puncta, $n = 25,053$ (38%), which was rejected based on their fluorescence emission of only a single channel (Supplementary Fig. 4c). We characterize this group as the "undetermined group" because each event can belong to: 1) the nonspecific binding of probes, 2) autofluorescent moieties, or 3) mRNA transcripts, which were not fully labeled with both dyes. For the first case, as previously characterized by several groups, nonspecific binding events is a common inherent issue with single-molecule FISH techniques which arises from the stochastic binding of DNA probes towards cellular components such as proteins, lipids, or nonspecific regions of RNA and follow a random distribution[14,20]. When combined with events which may be autofluorescence moieties (e.g., porphyrins, flavins), which can exist as isolated diffraction-limited structures and emit strong fluorescence in any particular single channel[38] or mRNA transcripts which were labeled with only one set of fluorophores, these groups represent a confounding issue for standard intensity-based measurements and analysis because they share similar SNR and intensities to real labeled puncta and cannot be differentiated without additional lengthy or complex techniques such as sample clearing or iterative-based labeling and imaging error correction[39]. Therefore, the main benefit of implementing the combinatorial encoded criteria is to ensure target detection fidelity by rejecting stochastic and nonspecific binding labeling events, as well as any event eliciting a lifetime signature that deviated from the utilized fluorophores. Finally, we also observed a relatively small group of puncta emitting fluorescent signal across more than two spectral channels but still eliciting the same spectral and lifetime signatures as the utilized fluorophores; $n = 2,439$. To characterize this population, we performed a simulation running 20,000 iterations of various puncta densities and fitted the corresponding exponential model that characterizes the probability of puncta overlap (described in Methods section and Supplementary Fig. 4a,b). We attained an interval for the fraction of lost puncta due to optical crowding ranging from 2.0 to 6.6%, which accounts for the 2,439 puncta (3.7% of the total detected puncta). We name this group the overlapping in Supplementary Fig. 4c.

The number of puncta detected of transcripts for each gene in each cell for the labeled 10-plex samples was plotted (Fig. 4h) and the mean number of detected puncta per cell split into the different genes classified using MOSAICA phasor analysis with combinatorial labeling. In comparison, we also show the MOSAICA pipeline results with the negative control sample obtaining counts of less than five per thousand mainly due to noise in the images (Fig. 4i). To validate these puncta count, we compared them to matching RNA-seq data from the same cell type with $n = 3$ experimental replicates (see replicate comparison in Supplementary Fig. 5). Shown in Fig. 4j is a scatter plot of the average mRNA puncta count for each cell plotted against the normalized counts from DESeq2 of our bulk RNA-sequencing data for each expressed gene. We obtained a Pearson correlation coefficient of $r = 0.96$, indicating a significant positive association between the two methods. Furthermore, to assess the rate of false positives and determine if one bright mRNA target could potentially be misidentified as another target, we repeated our experiment by leaving out probes for some expressed genes and then compared the detection rate of remaining targets with the 10-plex data. Specifically, we performed two additional experiments with an 8-plex, as well as two additional experiments with a 2-plex panel to compare the detected transcript abundance values and correlation coefficients against the 10-plex sample (Supplementary Fig. 6). We observed that there were no significant differences between these panel sizes in terms of target detection rate, indicating that target misidentification was not an issue for these panel sizes.

To further evaluate the detection efficiency, we performed benchmarking tests with our method against LGC Stellaris™ and RNAscope™ which are commercial gold standard FISH methods (Supplementary Fig. 7). Using the transcript of the housekeeping gene, POLR2A, as an exemplary target, we found a significant association between the number of detected puncta by our method and LGC Stellaris™ ($t$ test $p$ value $= 0.4$). When compared to RNASCOPE™, we observed that for this cell type and target, both our assays and LGC Stellaris™ did not correlate significantly ($p = 7.8 \times 10^{-4}$ and $p = 3.4 \times 10^{-4}$), indicating a discrepancy in detection efficiency between the two methods. We attribute this difference to MOSAICA and LGC Stellaris™ utilizing a direct labeling and amplification-free method while RNASCOPE™ utilizes a tyramide signal amplification reaction which generates thousands of fluorophore substrate per transcript and can lead to overlapping puncta or undercounting of detected puncta. Together, these data show MOSAICA can robustly detect target mRNAs of the broad dynamic range of expression levels from single digit to hundreds of copies per cell.

**Multiplexed mRNA analysis in clinical melanoma skin FFPE tissues**. We next investigated whether MOSAICA can provide multiplexed mRNA detection and phasor-based background correction and error detection to clinically relevant and challenging sample matrices. Assaying biomarkers in situ in tissue biopsies has great clinical values in disease diagnosis, prognosis, and stratification, including in oncology[40–42]. Specifically, we applied a mRNA panel consisting of KI67 (indicative of cell proliferation), POLR2A, BRCA1, MTOR, NCOA2, and NCOA3 to highly scattering and autofluorescent human melanoma skin biopsy FFPE tissues obtained from and characterized by the UCI Dermatopathology Center. Using the same probe design pipeline, primary probes were encoded with a combination of two fluorophores for the transcript of each gene to exhibit a unique fluorescent signature.

Figure 5b depicts a spectral image overlay (four fluorescent channels including DAPI) of the epidermis region of a labeled 6-plex skin tissue sample. Similarly, as in the previous section, the orthogonal lifetime image was attained after using phasor analysis to create the image depicted in Fig. 5c–e depict the merged composite

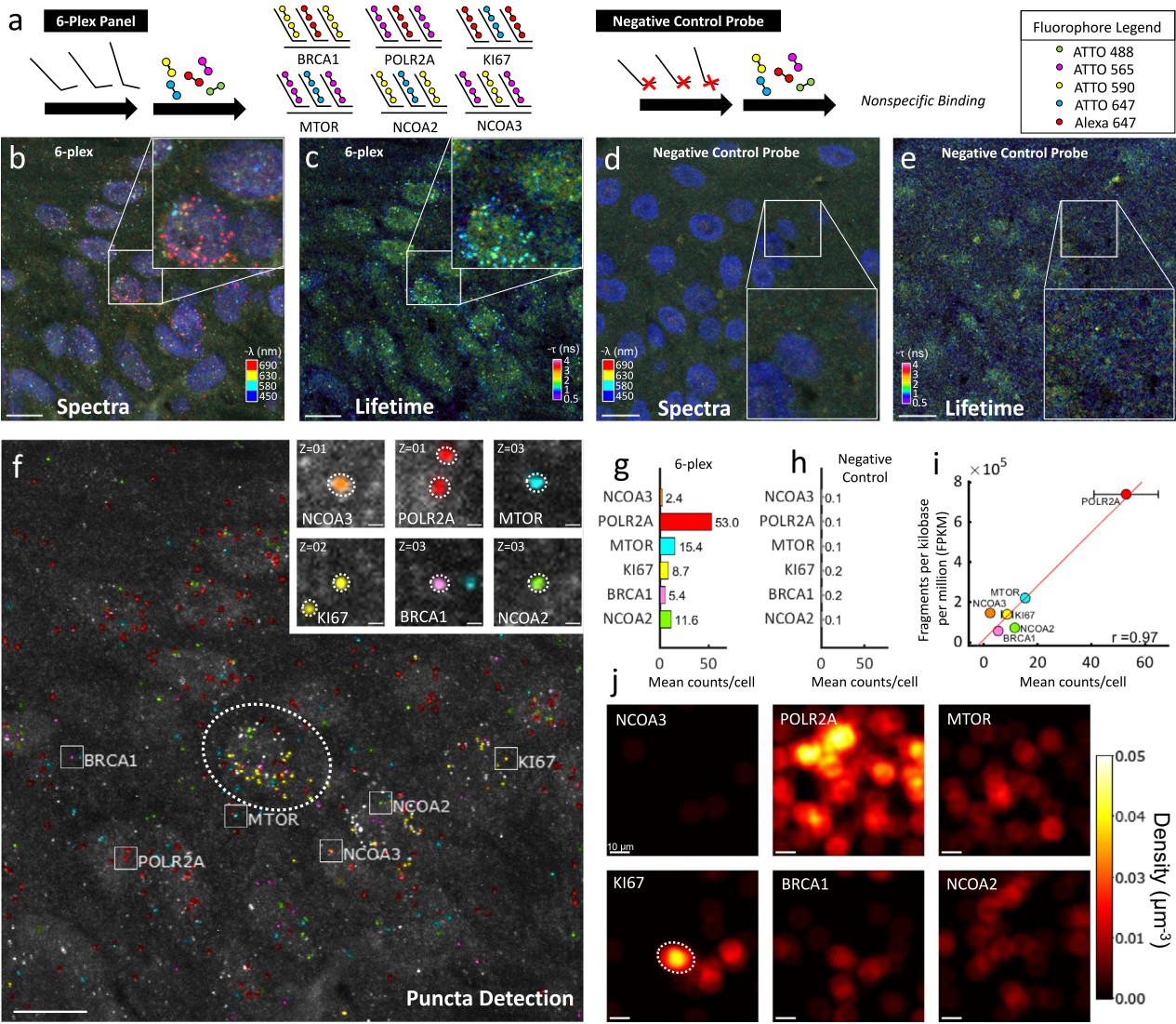

**Fig. 5 Multiplexed mRNA detection in epidermis region of human skin melanoma FFPE tissue. a** 6 different types of gene transcripts were labeled with primary probes followed by respective and complementary fluorescent secondary probes. Each transcript was labeled with a combination of two different fluorophores for six combinations. Negative control probes targeting transcripts not present in the sample were used with their respective secondary fluorophore probes. **b** Spectral image (max-projection in z) of a field of view of the labeled 6-plex sample (three channel pseudo coloring). **c** Lifetime image (max-projection in z) of a field of view of the labeled 6-plex sample (phasor projection on universal circle pseudo coloring). **d** Spectral image of the labeled negative control probe sample is depicted. **e** Lifetime image of the labeled negative control probe sample. **f** Final puncta detection of the 6-plex field of view after being processed in our analysis software showing highlighted example puncta of each target (insets, right). **g** Mean puncta counts per cell of transcripts for each gene in the 6-plex sample ($n=2$ experimental replicates, 174 cells). **h** Puncta count for the negative control probe sample ($n=2$ experimental replicates, 375 cells). **i** Correlation of detected puncta (mRNA puncta count) vs. bulk sequencing (fragments per kilobase per million) is shown for each target. **j** Transcript density in the field of view for each of the expressed genes reveals clustering of specific genes, as an example KI67 appears highly expressed in three cells, one of them marked with a dotted ellipse that corresponds to location in **f**). Scale bars 10 µm in large images and 1 µm in insets. Source data are provided as a Source Data file.

spectral and lifetime images of the corresponding negative probe sample also in the epidermis region. Figure 5f depicts the pseudo-colored puncta which were successfully classified and identified as their assigned mRNA markers. A representative inset image for each marker and its targeted detection is provided on the right. We observed that a population of puncta consisting of nonspecific, autofluorescent, or unknown sample artifacts rejected from analysis, (1,100) or 37.5% of the total detected puncta (2,934). In addition to this group, MOSAICA rejected a small group of puncta that emitted fluorescence in multiple spectral channels (62). This fraction (2.1%) is in concordance with the optical crowding range (2.0–6.6%) that our simulations and models predict (Supplementary Fig. 4). With conventional intensity-based measurements and analysis, both

contaminating groups are inherent image artifacts that compromise the integrity of puncta detection unless complicated quenching steps or additional rounds of stripping, hybridization, and imaging are utilized[14,43]. With MOSAICA, these contaminating artifacts can be accounted for with the integration of spectral, lifetime, and shape-fitting algorithms.

Figure 5g, h plots the total number of detected puncta for the labeled 6-plex sample and the negative control probe sample to highlight the final counts obtained using MOSAICA. To validate these puncta counts and their relative expressions, we examined the relationship between the decoded puncta with matching bulk RNA-sequencing obtained from The Cancer Genome Atlas (TCGA) database (see Methods section). Shown in Fig. 5i is a

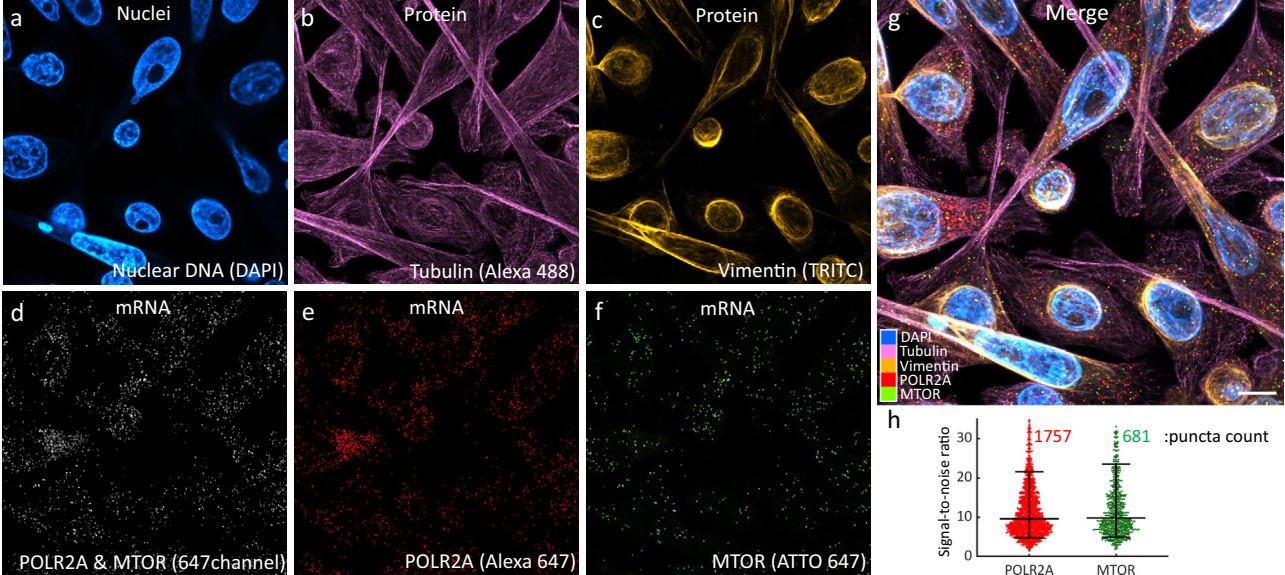

**Fig. 6 Simultaneous 4-plex co-detection of protein and mRNA in colorectal cancer SW480 cells. a** Intensity imaging showing nuclei labeled with DAPI. **b** Intensity image showing Tubulin protein labeled with Alexa488. **c** Intensity image showing Vimentin protein labeled with TRITC. **d** Intensity image at 647 nm showing mRNA targets, POLR2A and MTOR, which were further resolved by lifetime. **e** Unmixed lifetime image showing POLR2A puncta labeled with Alexa647. **f** Unmixed lifetime image showing mTOR puncta labeled with ATTO647. **g** Merged image of all channels. Scale bar is 10 μm. **h** Signal-to noise and puncta count analysis for the mRNA targets. Overlaid lines correspond to quantiles [10,50,90]%, *n*=1757 and *n*=681 transcripts respectively. Source data are provided as a Source Data file.

scatter plot of MOSAICA puncta count plotted against fragments per kilobase per million. We obtained a Pearson correlation of r=0.97 for this 6-plex sample, indicating a significant positive association between the two methods. We acknowledge that this strong correlation is particularly dependent on the presence of the highly abundant *POLR2A* expressed gene. The correlation for the other lower expressed targets excluding POLR2A is r=0.44 which, although still positive, is weaker. We attribute this discrepancy to preanalytical variables typically associated with FFPE sample preservation and pretreatment. For instance, there have been multiple studies, which documented increased variability in quantifying lowly expressed genes in FFPE tissues due to RNA degradation or cross-linking of proteins with nucleic acids[44–46]. Last, the density map of the detected transcripts provides a visual method to identify spatial localization of clusters of genes, such as *KI67* (indicative of proliferating tumor cells) being more prevalent in the dermis region while *POLR2A* is dispersed throughout the region (Fig. 5j). Overall, in situ profiling biomarkers, such as *KI67* and their spatial clustering can have diagnostic and prognostic values in malignant diseases and MOSAICA provides a robust platform to profile these markers[47].

**Simultaneous co-detection of protein and mRNA.** Spatial multiomics analysis including especially simultaneous detection of protein and transcript within the same sample can reveal the genotypic and phenotypic heterogeneity and provide enriched information for biology and disease diagnosis. As a pilot experiment to demonstrate MOSAICA's potential for multiomics profiling, we utilized MOSAICA to detect 2 protein targets, Tubulin and Vimentin, and 2 mRNA targets, *POLR2A* and *MTOR* in colorectal cancer SW480 cell culture samples (Fig. 6). After staining the sample with the primary antibodies, secondary antibodies were added to fluorescently label the protein targets. After protein labeling, we utilized the same probe design pipeline and labeling strategy for mRNA detection, primary probes were generated and hybridized to the sample after antibody staining. Corresponding secondary probes were hybridized. Figure 6a–f

depict the individual channels of the sample with Fig. 6g showing the merged channels of the 4-plex panel. As both *POLR2A* and *MTOR* are assigned to the 647 nm channel and cannot be separated spectrally (Fig. 6d), lifetime analysis is used to separate *POLR2A* (Fig. 6e) and *MTOR* (Fig. 6f). Signal-to-noise ratio measured as intensity of the detected puncta over intensity of the surrounding pixels was measured for the two mRNA targets (Fig. 6h). In summary, we have demonstrated MOSAICA as a potential spatial multiomics tool, which harmonizes sample treatment between both labeling processes. MOSAICA utilizes staining protocols with efficient target retrieval, blocking, and pretreatment steps where the viability and labeling of both target RNA sequence and protein markers were not compromised after each assay.

## Discussion
MOSAICA can fill a gap in the spatialomics field by offering both simplicity and multiplexing through direct in situ spatial analysis of a large number of biomarkers in a single round of staining and imaging (Supplementary Fig. 8). By contrast, conventional direct labeling approaches (e.g., RNAscope™, LGC Stellaris™, etc.) are limited to 3 or 4 targets. Other emerging spatial transcriptomics technologies such as seqFISH can offer greater multiplexing capabilities but requires many rounds of sample re-labeling, imaging, indexing, and error-prone image registration. The MOSAICA integrates both the spectral and lifetime dimensions and employs combinatorial target encoding, and phasor- and machine learning-based deconvolution to achieve high-plex analysis without sacrificing assay throughput. MOSAICA's error-detection feature can correct for stochastic nonspecific binding artifacts and autofluorescent moieties, inherent challenges associated with current intensity-based methods. MOSAICA's simple workflow can be particularly important in clinical settings where biopsy samples are limited in quantity and often not amenable for repeated processing. With respect to cost, MOSAICA uses inexpensive DNA primary probes and fluorescently labeled secondary probes which can be shared among

many different targets, reducing costs to several dollars per assay. Particularly compared to indirect spatial transcriptomic technologies that interrogate barcoded regions of interest (ROIs) with separate sequencing steps (e.g., 10x Genomics Visium, GeoMx® Digital Spatial Profiler), our direct, imaging-based approach can provide higher spatial resolution (single molecules or subcellular features), lower cost, simpler workflow, and potentially higher throughput (number of samples analyzed per time unit). Furthermore, our platform uses standard fluorescent probes and fluorescence microscopy and fluorescence imaging remains the most widely used technique in biological research. Several commercial instruments that can acquire both spectral and lifetime information are available including ISS FastFLIM, PicoQuant rapidFLIM, Leica SP8 FALCON, etc., and already exist in numerous shared facilities in industry and academia. Therefore, its minimal requirements of MOSAICA will permit quick and broad adoption in the scientific community.

MOSAICA holds great potential to broadly enable scientists and clinicians to better elucidate biological processes, and to develop precision diagnostics and therapies. Given that gene expression is heterogeneous and many different cell states can exist, one would need to assess multiple expressed genes within the same cell in situ. Therefore, it is anticipated that MOSAICA can enable spatiotemporal mapping in the attempt to construct 3D tissue cell atlas. In addition, MOSAICA can serve as a tool for targeted in situ validation of single-cell RNA sequencing data which reveal cell identities based on "differentially expressed genes" but are subjective, variable and error-prone. Furthermore, we aim to develop MOSAICA as a clinical companion diagnostic tool for stratified care. In particular, insights of the spatial organization and interactions between tumor cells, immune cells, and stromal components in tumor tissues can inform cancer diagnosis, prognosis, and patient stratification[48,49].

One common challenge in imaging-based spatialomics analysis is optical crowding which can limit both the number of molecules that can be detected and the detection efficiency and accuracy. For instance, as we scale up multiplexing capabilities by labeling more mRNA and proteins with additional fluorophores, more labeled targets and their fluorescent combinations will begin occupying the same voxel, leading to challenges in determining both how many targets there are as well as which type of targets are present within each voxel. We have modeled this phenomenon in Eq. 2 (Methods section) and plotted the results in Supplementary Fig. 4. Based on our estimates and in our current transcript density conditions, overlapping accounts for only around 6% of the detected puncta. We currently do not further resolve these cases and, instead, categorize them into the overlapping group, which do not contribute to total counts. In addition, we intend to further address these cases in the future using our multicomponent approach[36] to unmix spectral/lifetime components within a single voxel by means of higher harmonics of the phasor transform.

With respect to the crowding issue, the phasor analysis method has an additional limitation related to the use of the combinatorial technique. Even if the isolated fluorescent dyes are very far apart on the phasor space, their combinations fall inside the polygon as determined by the positions of the individual dyes. As one increases the number of dyes, the combinations start to overlap creating an ambiguity. For this reason, as we increase our multiplexed panel, our strategy is not only to employ labels which are distinctly separated by both spectral and lifetime properties but importantly to also use more combinations of different labels rather the same labels. The tradeoff between these two counteracting parameters is an exciting endeavor which we look forward to exploring as we progressively build up our repertoire of fluorophores.

Regarding the phasor population overlap, given the imaging settings we have used for the experiments in this paper, the signal-to-noise ratio produces gaussian phasor distributions with 99.7% of the pixels within 0.01 phasor units (6σ). Although the distributions tend to converge due to background autofluorescence, their small covariance matrices guarantee a high level of confidence in assigning each pixel to the correct cluster. As an example, the three gaussian distributions in Fig. 3h, have covariance matrices of (coordinate $S$ first, $G$ second) $\Sigma_1 = \begin{pmatrix} 12 & -5 \\ -5 & 89 \end{pmatrix} \times 10^{-5}$, $\Sigma_2 = \begin{pmatrix} 6 & -8 \\ -8 & 51 \end{pmatrix} \times 10^{-5}$ and $\Sigma_3 = \begin{pmatrix} 12 & -8 \\ -8 & 50 \end{pmatrix} \times 10^{-5}$ and mean coordinates of $\mu_1 = \begin{pmatrix} 0.26 \\ 0.40 \end{pmatrix}$, $\mu_2 = \begin{pmatrix} 0.25 \\ 0.49 \end{pmatrix}$ and $\mu_3 = \begin{pmatrix} 0.23 \\ 0.57 \end{pmatrix}$. With these values, the distance between the leftmost and right most cluster is of 0.17 phasor units, with the mean standard deviation from the covariance matrices being 400 times smaller at $3.65 \times 10^{-4}$. With numbers like these, we anticipate that our clustering technique can easily resolve even more challenging scenarios such as 6 lifetime phasor clusters and 10 spectral phasor clusters. As a result, our next immediate goal is to scale our multiplexing capability by detecting around 60 mRNA targets simultaneously with 12 different fluorophore species within the same sample. We aim to use seven spectrally distinct fluorophores and an additional five with overlapping spectra but are resolvable by lifetime. A combinatorial scheme of 12 choose two would yield 66 combinations. We could resolve these combinations using a seven-spectral channel instrument where five of the channels would present three populations in the lifetime phasor plot (one for each of the two probes with overlapping spectra in that channel plus the third being the combination of the two). Looking another step ahead, by implementing our recently developed 32-channel spectral-FLIM detector[34] which can provide 32 independent spectral sources with six lifetime clusters per channel, 192 different fluorophore species can be accessed to provide significantly higher plex detection capabilities.

In the future, we consider to expand our codebook by implementing a Förster resonance energy transfer (FRET)-based barcoding strategy[50] where different FRET fluorophore pairs and various distances between fluorophore donor and acceptor can be used to tune the combinatorial spectrum and lifetime readout. The FRET phenomena can also be used as an additional error-correction mechanism at the nanometer scale to potentially resolve multiple targets in the same voxel. Moreover, the current scanning confocal microscope implemented in MOSAICA can achieve high spatial resolution and z-sectioning but is limited by a relatively longer imaging time. As an example, each z-slice of our spectral-FLIM images (Fig. 4 and Fig. 5) took around 1.5 min ($1024 \times 1024$ images at 16 µs pixel dwell time, accumulating an average of 6 frames). However, we anticipate that this approach is compatible with any wide-field imaging technique as long as sufficient image pixel sizes, axial resolution, and photon counts are met. This can be accomplished with our recently developed camera-based light sheet imaging system[51] or a spinning disc confocal system equipped with a FLIM camera to greatly improve imaging throughput[52]. Indeed, MOSAICA is amenable for further integration with other imaging modalities, including expansion microscopy, super-resolution, and multiphoton imaging[53–55] to improve subcellular resolution and allow imaging large, scattering tissues. In addition, we will develop user-friendly image analysis software with capabilities enabling classification of single-cell phenotypes, spatial organization and neighborhood relationship among different cell types. Our puncta detection and classifier algorithm can be improved using convolutional neural networks

with clinical training sets to optimize biomarker detection accuracy and efficiency. Finally, we will develop high-plex protein detection component in our multiomics analysis using antibody-DNA conjugates where our combinatorial labeling and barcoding strategy can be used to scale up multiplexing.

## Methods

**Ethical statement**. We confirm this research complies with all relevant ethical regulations. The University of California, Irvine Institutional Review Board (IRB) approved this study for IRB exemption under protocol number HS# 2019–5054. All human melanoma FFPE cases were de-identified samples to the research team at all points and therefore considered exempt for participation consent by the IRB.

**Primary probe design**. A set of primary probes were designed for the transcript of each gene. A python code was used to rapidly design the primary probes while controlling various aspects of the probes such as GC content, length, spacing, melting temp, and prohibited sequences. To begin, probes are designed using exons within the coding sequence region. However, if that region does not provide over 40 probes, the exons from the coding and untranslated regions are used. The candidate probes are then aligned to the genome using Bowtie2, an NGS aligner, to determine if these probes are specific. Probes that are determined specific are then aligned to the RNA sequencing data on the UCSC Genome Browser, further eliminating probes that do not align to regions with an adequate number of reads. While mapping the probes to the genome on the UCSC Genome Browser, the probes are aligned with BLAT (BLAST-like alignment tool). A local BLAST query was run on the probes for the expressed genes in the panel to eliminate off-target hits. For this experiment, each expressed gene had the maximum number of probes that could be designed with our pipeline and requirements. The final primary probe design included two assigned readout sequences of the secondary probe with a "TTT" connector in between, another connector, then one of the probes specific for the transcript of that gene. The primary probes were ordered from Sigma Aldrich and pooled together for the transcript of each gene. The sequences of all probes used in this study are listed in Supplementary Data 1.

**Secondary probe design**. Secondary probe structures were based on the design from the Zhuang group[56]. In short, the 20-nt, three-letter readout sequences were designed by generating a random set of sequences with the per-base probability of 25% for A, 25% for T, and 50% for G. Sequences generated in this fashion can vary in their nucleotide content. To eliminate outlier sequences, only sequences with a GC content between 40% and 50% were kept. In addition, sequences with internal stretches of G longer than 3 nucleotides were removed to eliminate the presence of G-quadruplets, which can form secondary structures that inhibit synthesis and binding. To remove the possibility of significant cross-binding between these readout sequences, algorithms from previous reports were used to identify a subset of these sequences with no cross-homology regions longer than 11 contiguous bases[56]. Probes were then checked with BLAST to identify and eliminate sequences with contiguous homology regions longer than 11 nucleotides to the human transcriptome. From the readout sequences satisfying the above requirements, 16 were selected.

**Cell culture**. Human embryonic kidney (HEK293T) cells (632180; Takara) were cultured in DMEM (10-013-CV; Corning) supplemented with 10% FBS (1500-500; Seradigm) and 1% penicillin (25–512; GenClone). Human colorectal adenocarcinoma (SW480) cells were cultured in DMEM with high glucose (SH30081.02; HyClone) supplemented with 10% FBS (1500-500; Seradigm), 1x L-Glutamine (25–509; GenClone), and 1% penicillin (25–512; GenClone). SW480 cells were FACS-sorted based on surface marker ROBO-1, and ROBO + and ROBO- cells were used in Fig. 4 and Fig. 6, respectively. The cells were plated into 8-well chambers and then fixed. The eight-well plates (155409; Thermo Scientific) for HEK293-T and SW480 cells were coated with fibronectin bovine plasma (F1141-2MG; Sigma Aldrich) before seeding cells onto the 8-well plates. All cultures were grown at 37 °C with 5% $CO_2$.

**_mNeonGreen_ cell engineering**. A _mNeonGreen_ construct was transfected into HEK293T-X cells with FuGENE HD Transfection Reagent (E2311; Promega). The cells were then selected with puromycin (NC9138068; Invivogen) and Zeocin (AAJ67140XF; Alfa Aesar) 3 days after transfection.

**Preparation of fixed cells in cell chambers**. When the cells reached 70% confluency, cells were fixed for 30 min using 4% paraformaldehyde (15710; Electron Microscopy Science), then washed with PBS 3 times. The cells were then incubated with sodium borohydride (102894; MP Biomedicals) for 5 min and washed with PBS 3 times. 0.5% Triton X-100 (T8787-100ML; Sigma-Aldrich) in PBS was incubated in each well for 5 min and cells were washed with 2x SSCT (2x SSC with 0.1% TWEEN® 20 (P9416-100ML; Sigma-Aldrich). For storage, cells were left in 70% ethanol at 4 °C.

**Preparation of FFPE tissues**. The University of California, Irvine Institutional Review Board (IRB) approved this study for IRB exemption under protocol number HS# 2019–5054. All human melanoma cases were de-identified samples to the research team at all points and therefore considered exempt for participation consent by the IRB. Fully characterized human patient skin melanoma FFPE tissues with an immune cell score of brisk were obtained from the UCI dermatopathology center then sectioned to 5 μm slices using a rotary microtome, collected in a water bath at 35 °C, and mounted to positively charged Fisher super frost coated slides. The tissue sections were then baked at 60 °C for 1 h. For antigen unmasking, slides were deparaffinized, rehydrated then followed by target retrieval (with citrate buffer).

**Primary probe hybridization**. Blocking buffer containing 100 mg/ml Dextran sulfate sodium salt (D8906-100G; Sigma-Aldrich), 1 mg/ml Deoxyribonucleic acid from herring sperm (D3159-100G; Sigma-Aldrich), 0.01% Sodium Azide (S2002-100G; Sigma-Aldrich), 0.01% tween, and 15% ethylene carbonate (AC18410010; Fisher Scientific) in 2x sodium saline citrate (SSC) was added to the fixed cells or tissues and incubated at 60 °C for 8 min and then at 37 °C for 7 min. Following this preblock step, primary probes with 5 nM of each probe in blocking buffer were added to the samples and incubated at 60 °C for 30 min and then overnight at 37 °C.

**Secondary probe hybridization**. Once the primary probe solution is removed, the sample is washed with 2x Saline-Sodium Citrate Tween (SSCT) twice. Wash buffer (2xSSCT with 10% ethylene carbonate) is used for 3 washes and incubated in 60 °C for 5 min each time. Blocking buffer is added and incubated at room temperature for 5 min. The sample is then incubated in a solution with 5 nM of the secondary probes in blocking buffer at 37 °C for an hour. The sample is washed with 2x SSCT twice before using wash buffer to wash 3 times and incubated in 42 °C for 5 min each time. For the first wash, 10 mg/mL Hoechst (H3570; Invitrogen) is diluted 1:1000 in PBS and added to cells. Later, the wash buffer is then removed and replaced with glycerol mounting media and ready for imaging.

**Codetection of protein and mRNA**. Prior to mRNA labeling, fixed SW480 cells were blocked with 1% Bovine Serum Albumin (RLBSA50; VWR), 0.1% TWEEN® 20, 1:1,000 Sodium Azide, 0.2 U/ml Protector RNase inhibitor (3335399001; Sigma-Aldrich), and 1 mM DTT in RNAse-free PBS (AM9625; Life Technologies) for 30 min at room temperature. These cells were then washed 3 times with 0.1% TWEEN® 20 in RNAse-free PBS for 5 min each wash at room temperature. Antibody solutions containing 1:1,000 Mouse anti-Tubulin (3873BF; Cell Signaling) and 1:200 Rabbit anti-Vimentin (5741BF; Cell Signaling) in the same blocking buffer were subsequently added to the samples and incubated overnight at 4 °C. Following 3 additional washes with 0.1% TWEEN® 20 in RNAse-free PBS for 5 min each at room temperature, antibody solutions containing fluorescently labeled 1:200 Donkey anti-Mouse Alexa-488 (R37114; Fisher Scientific) and 1:200 Donkey anti-Rabbit TRITC (711-025-152; Jackson Laboratories) in the same blocking buffer were added at room temperature for 1 h. After 3 washes with RNAse-free PBS with 0.1% TWEEN® 20 for 10 min each wash at room temperature, 4% PFA was added for 15 min at room temperature. These cells were then washed 3 times with 0.1% TWEEN® 20 in PBS at room temperature for 5 min. For mRNA labeling, the previously described methods regarding primary and secondary probe hybridization were utilized.

**LGC Stellaris ™**. LGC Stellaris[TM] RNA FISH probes (Biosearch Technologies, CA, USA) were used, with 48 × 20 mer fluorophore-conjugated oligos tiling the length of the target transcript. The _POLR2A_ probe set were supplied as predesigned controls conjugated to Quasar 570 fluorophores. Labeling/staining was carried out as described in the LGC Stellaris[TM] protocol for adherent mammalian cells. The POLR2A probe sets were used at 50 nM.

**RNAscope[TM]**. The FFPE tissue sections were deparaffinized before endogenous peroxidase activity was quenched with hydrogen peroxide. Target retrieval was then performed, followed by protease plus treatment. The fixed cells pretreatment included treatment with hydrogen peroxide and protease 3. The RNAscope[TM] assay was then performed using the RNAscope[TM] Multiplex Fluorescent V2 kit and Akoya Cy5 TSA fluorophore. The positive control (_POLR2A_) and negative control (_dapB_) were in C1.

**Microscopy Imaging**. Our samples can be imaged with any instrument provided that it has spectral and lifetime acquisition capabilities. Our measurements were taken on three separate instruments, a wide-field and two confocal microscopes. A generic spectral-FLIM scanning confocal instrument setup is depicted in Supplementary Fig. 9.

For validation of fluorophores and their spectral and lifetime signatures, measurements were taken on a 2-channel ISS Alba5 STED platform. This system is equipped with a pulsed white laser (NKT SuperK EXTREME) system where the excitation wavelength(s) can be selected with an acousto-optic tunable filter (NKT SuperK SELECT). Single photons were detected with avalanche photodiode

detectors (Excelitas Technologies) and their arrival times with respect to the stimulating frequency (78 MHz) were measured with a FPGA-based electronic board (ISS FastFLIM). Imaging was achieved by fast beam scanning with galvo mirrors and 3D stacks of images were acquired with a z-piezo mount on the objective.

For measurements of multiplexed/combinatorial labeling and detection experiments (Fig. 4 and Fig. 5), we utilized a Leica SP8 with the Falcon module. This platform employs a white light laser and an acoustic optic beam splitter dichroic, and the Leica hybrid detectors with excitation band selectable by means of a prism. 3D measurements of cells and tissue samples were taken with a 100x plan apochromat oil objective with a numerical aperture of 1.40, yielding images with an x-y resolution of 100 nm and z-spacing of 500 nm.

For epifluorescence measurements (Supplementary Fig. 2 and Supplementary Fig. 3), images of labeled mRNA transcripts were taken on an inverted Ti-E using a 100× plan apochromat oil objective with a numerical aperture of 1.40. Samples were illuminated with a Spectra-X (Lumencor) LED light source at the 395 nm, 470 nm, 555 nm and/or 640 nm excitation wavelengths. Images were acquired with an Andor Zyla 4.2 sCMOS camera at 4 K resolution with 6.5 μm pixels.

**Image Processing**. A custom set of scripts were written in MATLAB to process the acquired image stacks, identify individual transcripts and assign each of them to each gene expression target. After reconstructing the images out of the digital list of photons, the analysis runs in parallel a 3D blob detection pipeline on the intensity image stacks to identify each transcript and on the other a clustering pipeline on the phasor-transformed lifetime/spectral phasor data to detect distinct spectral/lifetime populations. A classifier then assigns pixels as belonging to a particular expressed gene. The whole pipeline is depicted in Supplementary Fig. 10.

Briefly, the intensity 3D stacks are run through a blob detection algorithm that was developed in order to identify each transcript. The images can be seen as a 3D space where the transcripts appear as spherically symmetric locations with a radial increase in intensity, namely puncta. The algorithm first computes the low-frequency background noise by means of a median filter with a kernel 10 times the size of the expected puncta (the diffraction limit of the instrument, in our case around 250 nm). This low-frequency background is subtracted from the high-pass filtered data obtained by convolving by a gaussian filter of the expected size of the puncta. This on one hand enhances the puncta in the image by giving a prominence value at each pixel with respect to the surrounding regions and on the other suppresses noise in the images. A search for local maxima is performed by finding the locations where the gradient goes to zero and the divergence of the gradient is negative. Once the centers in the 3D coordinate space are obtained the size, absolute brightness and prominence of each puncta is measured.

In parallel, the raw photon counts are used to construct the photon arrival time histogram and photon spectral histogram at each pixel. Phasor transforms are applied to each pixel in each image of the 3D stack in order to construct the stacks' phasor plot. This phasor data is in general a 4-dimensional, each pixel in the intensity image has four additional coordinates; two for the spectral phasor transform plus two for the lifetime phasor transform. The phasor coordinates are clustered using Gaussian Mixture Models[57]. We used an initial experiment tagging the transcripts of housekeeping genes in order to guarantee that all expected populations were present and we trained the Gaussian Mixture Model using this initial experiment. This pretrained model is then applied to the new sets of data in order to classify each pixel into one of the clusters allowing for the presence of empty clusters. The number of clusters N intuitively should be the number of distinct fluorescent probes or different combinations of probes used to tag the sample, but one must allow for additional populations in the sample, e.g., autofluorescent species. For this reason, in the training of the Gaussian Mixture Models we allowed for one additional cluster to account for autofluorescence and noise.

Finally, by computing the mean phasor coordinates of the pixels within each detected puncta, we can compute the phasor position of each puncta and assign a gene expression label to it by a priori knowing the expected positions of each combination of probes depending on the spectra and lifetime of the probes.

To obtain the number of counts per cell, DAPI image stacks are segmented by means of simple thresholding, estimating the threshold value by hard-splitting of the histogram of photon counts in the channel. The 3D segmented nuclei are then iteratively grown by convolution by a minimal 3x3x3 kernel. This convolution is applied at each pixel of the edge of the segmented volume until no available space is left between the segmented volumes. This yields a division of the imaged volume into polyhedra where each face is exactly the plane bisecting the two closest nuclei edges. This process is analogous to a Voronoi tessellation using the surface of the nuclei instead of points.

In the cell culture experiments, to normalize by cell volume, we used a normalized mean cell volume of 3000 μm³ since a cell marker was not utilized and the imaged volume thickness (5 μm) was less than the actual cell thickness. To obtain mean counts per cell, total detected puncta counts was divided into the total imaged cellular volume and then multiplied by an estimated mean cell volume of 3000 μm³. The total cellular volume was obtained by an intensity threshold segmentation of the background cellular autofluorescence over the gaps between cells.

**Simulations**. In order to test the detection and classification pipeline, we wrote a set of scripts to simulate spectral/lifetime data which provided a ground truth towards detection and accurate classification debugging. This data generation script allows randomly distributing N diffraction-limited transcripts in an arbitrarily big 3-dimensional space, each with a gaussian intensity profile. We simulated our transcript gaussian profile with a X-Y standard deviation of 200 nm and a Z standard deviation of 500 nm, a peak intensity of 1 ± 0.3 (the intensity becomes relevant when simulating background noise). In the simulation run that we used to test the crowding limitations of the system we simulated tagging the transcripts of genes with couples out of a total of 12 fluorescent probes; 4 distinct spectral probes and 3 distinct lifetimes in each, yielding a total of $\binom{12}{2} = 66$ possible expressed genes.

We generated the simulated images in a cubic space of $10 \times 10 \times 10$ μm, discretized as an image stack of 33 images of $1000 \times 1000$ pixels (yielding a voxel resolution of $100 \times 100 \times 300$ nm). This volume was generated containing increasing densities of transcripts, ranging from a single transcript of each gene (66 transcripts) up to 2000 transcripts of each gene (132k transcripts) and for each possible value of density a total of 10 iterations each time. These 20k simulated image stack sets were individually processed by our image processing pipeline and the transcript position and labelling obtained by the pipeline was compared to the known ground truth of the generated data. This simulation provided a benchmark of the density limitations of the method but at the same time giving an idea of the underestimation of the number of transcripts as a function of local density. The simulations allowed us to model the estimated number of overlapping transcripts as a function of density.

A similar set of simulations was run by emulating the conditions in the 10-plex experiment (Fig. 4) where the transcripts of genes are tagged with combinations of two out of five probes. The 20k iterations for different densities allowed to plot the density of the classification obtained after detection compared to the real number of transcripts in the simulations. This simulation was fit to the probabilistic model obtained from calculating the number of transcripts that are not overlapped in space (see next section), from which the true number of puncta was extracted (see Supplementary Fig. 4).

**Overlapping probability**. The fraction of puncta that do not overlap with any other puncta depends on the total number of puncta present in the volume of study and the relative proportion between said total volume and the volume of each individual puncta. The following expression is obtained as the product of N-1 times the fraction of available space having removed the volume occupied by one transcript:

$$\frac{n}{N} = \left(1 - \frac{v_i}{V_T}\right)^{N-1} \tag{2}$$

where $n$ is the number of isolated puncta, $N$ is the total number of puncta, $v_i$ is the volume of each puncta and $V_T$ is the total volume (simulated or scanned). The real number of transcripts N cannot be analytically isolated from the previous equation, but one can graphically obtain it. Due to the fact that the transcripts are sub-diffraction limit, the value of $v_i$ is simply the volume of the point spread function of the instrument. Using the detected number of counts in an experiment $n$=13.5k and the estimated total imaged cellular volume of 68 kμm³, both obtained from two image stacks shown in Fig. 4, we proceeded to estimate the real number of transcripts present in the sample using the previous expression. Assuming an interval of possible volumes for the transcripts (instrument PSF) between 0.1 and 0.3 μm³ we obtained an estimated percentage of overlapping puncta in the interval [2.0, 6.6]%. This range of values is in agreement with the number of puncta that we detected in more than two channels in the 10-plex experiments (3.7%) and in the tissue experiments (2.1%). See Supplementary Fig. 4 for additional details such as expression (2) plotted as a function of the density of transcripts.

**Sequencing Data**. Colorectal cancer SW480 cell bulk RNA sequencing (unpublished data) was analyzed with DESeq2. Average expression is then obtained for comparison to the MOSAICA puncta count for each expressed gene. For the human skin melanoma FFPE tissue, the patient sample did not have corresponding sequencing data. RNA sequencing data were obtained from publicly available data from The Cancer Genome Atlas (TCGA), available on the National Cancer Institute (NCI) Genomic Data Commons (GDC) data portal, from 5 human skin melanoma FFPE biopsy thigh punch samples [Entity ID: TCGA-EE-A2GO-06A-11R-A18S-07, TCGA-EE-A20C-06A-11R-A18S-07, TCGA-YG-AA3N-01A-11R-A38C-07, TCGA-DA-A95Z-06A-11R-A37K-07, TCGA-GN-A26C-01A-11R-A18T-07]. The sequencing data were analyzed with HTseq and normalized for sequencing depth and gene length using Fragments Per Kilobase Million. The average of the 5 patient samples for each transcript were used for correlation graphs with MOSAICA puncta count.

**Experimental replicates and reproducibility**. Figure 3 is a conceptual figure and a single experiment was used as an example without replicates. For the 10-plex cell culture experiments (Fig. 4), we ran 3 experimental replicates from which we imaged 6 fields of view ($100 \times 100 \times 5$ μm each), with 364 cells in total. In these

image stacks, a total of 65,562 puncta were detected where 38,056 were assigned and 27,506 were unassigned to a target. The unassigned counts were further categorized based on assumed overlapping errors (2439) or as undetermined counts (25,053). In the associated negative controls, a total of three experiments were performed, of which we imaged 4 fields of view containing 189 cells total. In these experiments, a total of 2034 puncta were detected, of which 61 were classified as targeted expressed genes due to the expected spectral and lifetime signature and 1959 were classified as undetermined.

The tissue experiments with a 6-plex gene expression panel were replicated a total of 2 times yielding 2 fields of view of $130 \times 130 \times 3\ \mu m$ each, together containing 174 cells (Fig. 5). A total of 2934 puncta were detected of which 1770 were assigned to a target and 1164 were unassigned, the latter group divided into 62 puncta unassigned due to overlap and 1100 labelled as undetermined. In the associated negative controls, we ran a total of 3 experiments yielding 3 fields of view and 375 cells. In these fields of view, 390 puncta were detected, of which 43 were assigned to the transcripts of targeted genes. Of the other 347, only 4 were assigned to overlap and 339 to undetermined. The protein-mRNA codetection experiment in Fig. 6 is a pilot experiment for demonstration purpose and there is no replicate for it. Additional 8-plex and 2-plex experiments were performed on cell cultures, two replicates each, yielding a total of 143 and 130 profiled cells, respectively. Quantification of the experimental replicates by means of cross correlation is presented in Supplementary Fig. 5 and Supplementary Fig. 6.

**Statistical Analysis**. When comparing distributions of puncta counts, signal-to-noise ratios, and intensity values, Student (two-sided) t-tests were performed against the probability that the measured distributions belong to distributions with equal means. The reported probability values in the figures are symbolized with (* for $p < 0.05$, ** for $p < 0.01$, *** for $p < 10^{-3}$, and **** for $p < 10^{-4}$). Pearson correlation coefficient was computed to determine the correlation between the average expression level to the puncta count of each transcript and to compare within replicates of same experiments. For comparison of 2-plex gene expression counts, we implemented a binomial test where we used the obtained proportion of counts from the 10-plex experiments as the reference probability (Supplementary Fig. 6).

**Reporting summary**. Further information on research design is available in the Nature Research Reporting Summary linked to this article.

## Data availability

Source data containing underlying data points are provided with this paper for Figs. 3, 4, 5, and 6 and supplementary Figs. 2-7. All the raw data that were used for all the experiments including images, and the processed figure source data that are portrayed in each panel have also been deposited in the public repository Figshare[58]: [https://doi.org/10.6084/m9.figshare.17072390.v5]. A working example (used in Fig. 4) is included with the software package in the Code Availability section. Probe sequences used for labelling are included in the supplementary material section (Supplementary Data 1). RNA sequencing data were obtained from publicly available data from The Cancer Genome Atlas (TCGA), available on the National Cancer Institute (NCI) Genomic Data Commons (GDC) data portal [https://gdc.cancer.gov/]. Entity IDs: TCGA-EE-A2GO-06A-11R-A18S-07, TCGA-EE-A20C-06A-11R-A18S-07, TCGA-YG-AA3N-01A-11R-A38C-07, TCGA-DA-A95Z-06A-11R-A37K-07, TCGA-GN-A26C-01A-11R-A18T-07. Source data are provided with this paper.

## Code availability

Scripts and algorithms used for the image manipulation, puncta detection and gene classification have been uploaded to the public repository Figshare[59]: [https://doi.org/10.6084/m9.figshare.14810820.v4].

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

## Acknowledgements

We thank Dr. Arthur Lander and the UCI Cancer Systems Biology U54 center for their scientific inputs, Dr. Delia Tifrea, Dr. Jeffrey Kim and the UC Irvine Experimental Tissue Resource (ETR) for their help on tissue preparation and pathological characterization, Fairlie Reese for advice about sequencing data processing, Amber Habowski for help processing the bulk RNA sequencing data, and UCLA's California NanoSystems Institute and UCI's Beckman Laser Institute for their microscope support. Some figures were partly created using BioRender.com. This work was funded by a U54CA217378 grant to the UCI Cancer Systems Biology Center (CaSB@UCI), NIH/NIAMS P30AR075047 Skin Biology Resource Center grant, a seed grant from UCI Precision Health through Artificial Intelligence Initiative, a P30CA062203 cancer center support grant to the UCI Chao Family Comprehensive Cancer Center and UCI's Genomics and High Throughput Facility and Experimental Tissue Resource (ETR), NIH/NIGMS R21GM135493 to P.N.H. and NIH grant P41GM103540 to E.G. Some imaging was performed on the SP8-Falcon at the Advanced Light Microscopy/Spectroscopy Laboratory and the Leica Microsystems Center of Excellence at the California NanoSystems Institute at UCLA with funding support from NIH Shared Instrumentation Grant S10OD025017 and NSF Major Research Instrumentation grant CHE-0722519. L.H. and T.V. were supported by NSF GRFP (DGE-1839285). A.V. was supported by the Balsells Fellowship, Generalitat de Catalunya. J.G. was supported by a UCI Immunology NIH T32 training grant AI 060573 and National Institute of Neurological Disorders and Stroke (NINDS/NIH) Training Grant NS082174.

## Author contributions

T.V., A.V., and J.G. designed, conducted and analyzed the experiments. T.V., A.V., J.G., and W.Z. wrote the manuscript. K.L., Q.X., J.F., J.Z., and C.D. conducted experiments. J.S., L.H., J.W., M.L.W., A.G., P.H., and E.G. provided technical support and consulted on the study. E.G. and W.Z. designed and directed the project.

## Competing interests

T.V. and A.V. are now Arvetas Biosciences Inc employees. W.Z. is a cofounder of Velox Biosystems Inc., Amberstone Biosciences Inc., and Arvetas Biosciences Inc. P.N.H. holds a part-time position at Amberstone Biosciences Inc. A.G. is a founder of Alyra therapeutics. The remaining authors declare no competing interests.
