## [Peer Review File · Nature Communications]

Spatial transcriptomics using combinatorial fluorescence spectral and lifetime encoding, imaging and analysisReviewers' Comments:

Reviewer #1:

Remarks to the Author:

Vu and colleagues present in this manuscript a novel approach to the multiplexed imaging of RNAs and proteins, which they term Multi Omic Single-scan Assay with Integrated Combinatorial Analysis (MOSAICA). In MOSAICA, the authors leverage spectrally resolved fluorescence lifetime imaging to extend the number of RNA molecules that can be identified with combinatorial barcoding beyond that which could be identified with fluorophore color alone. The authors label individual RNA molecules using standard single-molecule fluorescence in situ hybridization (FISH) approaches and in the process assign to each molecule a unique barcode. In turn, this barcode is comprised of the unique combination of two of multiple different fluorescent dyes, which can overlap in color or fluorescent lifetime but not both. By leveraging color- and lifetime-resolved measurements, the authors can then distinguish the specific dyes associated with each RNA and, thus, identify the RNA molecule.

The authors validate their approach by measuring 10 genes in cell culture, where they show relatively low rates of false measurements in companion controls and strong correlation between the RNA abundances determined via MOSAICA and via bulk RNA-sequencing. They also validate MOSAICA by measuring 6 genes in formalin-fixed paraffin-embedded (FFPE) melanoma samples. Again, they show relatively low rates of false detection and an overall agreement between abundances determined via MOSAICA and bulk RNA-sequencing.

Image-based approaches to spatial transcriptomics represent an exciting and growing set of tools, and, in this respect, the introduction of time-resolved imaging into this toolbox represents a novel and important contribution. For this reason, we are supportive of the publication of this manuscript; however, there are a few issues that we feel should be addressed in revision to both fix a few minor errors and better benchmark the performance of this technique.

First, while the authors have provided a well described set of proof-of-principle and benchmarking experiments, there are several aspects of MOSAICA that are not described, measured, or quantified that are likely to be of interest to anyone considering adopting their technique.

- 1) Could the authors provide measures of reproducibility in their measurements? For example, how well do the abundance values determined via MOSAICA in replicate measurement in cell culture or in the melanoma sample correlate with one another? At least two replicate measurements for each experiment should be presented for a new technique.
- 2) The authors provide a nice benchmark measurement for the rate of false positives in which they leverage FISH probes that do not target RNAs in their samples. However, the false positive rate measured in samples in which there are no bright FISH signals may not be representative of the real false positive rates in a full MOSAICA measurement. For example, such measurements may not capture the rate at which one bright RNA is misidentified as another bright RNA. Could the authors provide a measure of their false positive rate that captures this aspect? A straightforward measurement of this property should be possible with their existing reagents. For example, they could repeat the cell culture experiments but leave out the probes to one or two genes. The rate at which these genes are detected would then provide this additional false positive rate. Such a measurement would have the added benefit of more closely reflecting the standard approach for estimating false positives in other image-based transcriptomic techniques, i.e., MERFISH, seqFISH, starMAP etc.
- 3) The authors contrast MOSAICA to other image-based techniques and highlight the potential for MOSAICA to be faster. However, these other techniques, MERFISH/seqFISH/starMAP, are compatible with wide-field imaging whereas MOSAICA is likely constrained to point scanning. The authors should provide some information on how long it takes to collect an image of a given size so that readers can better judge the potential throughput of this technique.
- 4) It may be worthwhile for the authors to note that their high correlation coefficient between bulk RNA-seq and MOSAICA for their melanoma measurements would appear to be dominated by the measurement of one, highly abundant gene: POLR2A. The correlation within the other genes is likely far more modest. FFPE can be challenging sample source for FISH measurements, so this issue is likely not reflective of a problem with MOSAICA. Nonetheless, the authors may wish to better explain this high correlation coefficient.

5) Finally, for their next step, the authors aim to expand MOSAICA to a 60-plex scheme that can detect 1770 mRNA species at once. However, the multitude of fluorophores will likely cause the spectral and lifetime phasor spaces to become overcrowded, thus increasing measurement error rates. If the authors would like to propose that 60-plex (or higher multiplexing in general) is possible, it would be helpful to provide some rationale as to what number of distinct fluorophores could be observed and the practical and technical limitations on this number.

Second, while we understand the desire to set MOSAICA in the most favorable light relative to other techniques, we have several concerns with the way in which prior work is described. There is a growing array of techniques that fall into the category of spatially resolved transcriptomics, yet the description and grouping of these techniques is not always accurate, and fundamental techniques are not mentioned.

1) The authors group together single-molecule imaging techniques, e.g., MERFISH, seqFISH, RNAscope, and SABER, with spatial capture methods, e.g., GeoMx, slide-seq. These technologies are not easily compared both in resolution, performance, and technical requirements. Perhaps it would be clearer to simply focus the comparison on imaging-based techniques, which is the category in which MOSAICA clearly falls.

2) Moreover, the authors call GeoMx and slide-seq in-situ sequencing techniques, yet this definition is not the standard categorization in the field. These techniques do not sequence molecules in situ but rather incorporate spatial barcodes with cDNA created from captured molecules which are then removed and sequenced ex situ to reconstruct molecular identity and location. The more standardly defined in situ sequencing techniques by Nilsson (ISS), Church (FISSEQ), Deisseroth (starMAP), and Boyden (ExSeq) are not mentioned, and if the authors feel it is important to contrast MOSAICA with other approaches for multiplexed single-RNA-molecule identification, they should probably mention these methods.

3) It is not accurate to describe all multiplexed FISH techniques as involving stripping or causing damage to tissues. Some techniques, such as MERFISH and a few newer implementations of seqFISH, do not remove secondary (readout) probes from primary (encoding) probes but rather chemically cleave or inactivate fluorophores. The former approach is proven to be quite gentle, and samples are stable through tens of rounds of this process, in terms of both tissue structure and mRNA quality. A major technical advantage of MOSAICA is the fact that repetitive sample processing is not needed, but it is important to note that methods exist for this repetitive processing to maintain sample integrity. In short, sample degradation is not a drawback of all repetitive processing methods.

4) There are several places in the manuscript where concrete numbers are provided for potential improvement of MOSAICA over other techniques (i.e., "For a 10-plex panel, this drastically shortens assay time by at least of factor of 4-5x and scales up even more as the panel becomes more multiplexed...") In situations such as this one, these arguments should be supported with estimates drawn from the literature for the performance of existing techniques and measurements of the performance of MOSAICA. For example, as a point-scanning method it is possible that image acquisition is much slower than wide-field methods, so the enhancement of speed that comes with the removal of repetitive staining is not obvious.

5) In several locations it is stated that MOSAICA is an error-correcting technique, yet that statement does not appear to be true. Error correction formally requires that a measurement that is incorrect can be associated with a correct barcode. Removal of noise sources or background are not considered error correction. The encoding scheme utilized in this work is a n-choose-k scheme in which k is 2. This scheme can detect errors, i.e., any spot that only has one measured dye instead of two clearly had a measurement error, but it does not have a way to predict the fluorophore that was lost in this case. In addition, if an mRNA spot had two fluorophores and one of them is incorrectly detected as a wrong dye, this coding scheme will not detect any error but simply decode the mRNA as an incorrect species. Thus, these codes have limited error-detecting capacity but not error correcting capacity.

Reviewer #2:

Remarks to the Author:

In the manuscript "Spatial transcriptomics using combinatorial fluorescence spectral and lifetime encoding, imaging and analysis" from Vu, Vallmitjana, Gu et al, the authors describe a new

method combining in situ labeling of mRNA and protein markers with spectral and lifetime imaging and machine learning.

The novelty of the work is the combination of labels with different lifetimes/spectra with dual-domain spectra-FLIM imaging (application of the s-FLIM technique from the Gratton group by Scipioni et al (Nature Methods)) and machine learning phasor clustering to spatially multiplex transcriptomics data.

There are limited solutions for performing parallel multi-label spatial transcriptomics, mostly owing to the challenges imposed by the labels' spectral emission overlap. Here the authors exploit imaging of fluorescent spectral and lifetime domain to provide high-throughput information-rich imaging data.

The authors claim that the MOSAICA approach:

1. Can spatially reveal and visualize the presence, identity, expression level, location, distribution and heterogeneity of each target mRNA in the 3D context
2. Can perform 10-plex imaging with error -correction and autofluorescence removal

The first claim is supported with examples and analysis of experimental data. The second claim is supported by comparison of real data with "artificially filtered" data, simulations and comparison with standard analysis approaches.

In all cases, the authors showcase the capabilities of MOSAICA in multiplexing, providing some quantitative parameters. According to the experimental data the performance of MOSAICA is promising, with 4-5x assay time shortening and 10-20x cost reduction.

In summary, the paper presents a novel technique and, if the claims stated by the authors are confirmed (see my major comments below), it represents an original and significant technical advance for spatial transcriptomics. I think it is of interest and of potential impact for several applications, especially for diagnostics.

For these reasons I think that the paper is of great interest for Nature Communications and well-worth of publication on this journal.

However, there are some fundamental points that must be addressed to understand the true advantages of the proposed method. The quantification of performance of the technique is partially present in the manuscript, although scattered between main text and supplementary. The performance/repeatability of the combinatorial labels in the aspect of instrumental noises is lightly described and not well highlighted for a broad viewership.

Finally, the manuscript writing is acceptable and, with minimal improvements, can cater to the broad audience of Nature Communications.

I ask the authors to address the following concerns.

Major comments

1. The authors lightly describe the performance of the combinatorial labeling in the aspects of label-implementation, detection and noises of the system. The Supplementary Figure 4 reports an informative description, and one of the few quantifications, of the performance of the MOSAICA (Supp. Figure 4 C,D). In this figure 56% of puncta are estimated to be assigned correctly, 37% is undetermined and 6% is overlapping. The main text only references the 6% lost due to overlap. The 37% undetermined should be reported more clearly and discussed in the main text as most readers will be interested in this information.
2. In this work, labels with different lifetime/spectra are combined ad hoc to fill a discretized matrix in the lifetime/spectra domains (as shown in Figure 1 F). The repeatability of this combinatorial label will determine the precision of the multiplexing. Example: the combination that falls in Target6 (Figure 1F) needs to be sufficiently different from the combination that falls in Target 7. Did the authors characterize the uniformity of a single combinatorial label? Example: measuring only the combination of lifetime/spectra that provides Target6 (Figure 1F), what is the average lifetime/spectra and their respective standard deviation? The broadness of this standard deviation will affect both accuracy of measurement and the total multiplexing capability. The authors should discuss this in the main text.
3. The effects of point 2 above are amplified by the presence of noise in the instrument. Measurement of 1 spectrum (or lifetime) on phasor produces a gaussian distributed cluster. Provided n combination of labels, the tails of these clusters will increasingly overlap with the number of combinatorial labels. The authors pick labels that are well separated on phasor to minimize this overlap, however supp. Figure 4 does report some information loss. Based on these negatively affecting components, what is the maximum number of plex that MOSAICA could

reliably identify? The authors should discuss this in the main text.

4. How do the authors account for when two combinatorial labels fall within the same pixel? E.g. looking at figure 3 H, if one magenta label and one green label fall within the same pixel the apparent signal will be orange, providing an incorrect assignment of the label. This problem increases with the number of possible target genes (equation 1), quadratically. The authors should discuss this in the main text.

Minor comments:

5. In introduction section, line 122 "we were able to discriminate a large repertoire of spectral and lifetime components", providing an actual number would highly improve the impact of the work.

6. Line 151 "imaged using a confocal microscope (Fig. 1D) equipped with spectral and lifetime capabilities", if this is the same instrument described by Scipioni et al (Nat. Methods 2021), please reference the work, else if this is the SP8-falcon mentioned in Methods section, add "commercial unit" or similar.

Reviewer #1

Vu and colleagues present in this manuscript a novel approach to the multiplexed imaging of RNAs and proteins, which they term Multi Omic Single-scan Assay with Integrated Combinatorial Analysis (MOSAICA). In MOSAICA, the authors leverage spectrally resolved fluorescence lifetime imaging to extend the number of RNA molecules that can be identified with combinatorial barcoding beyond that which could be identified with fluorophore color alone. The authors label individual RNA molecules using standard single-molecule fluorescence in situ hybridization (FISH) approaches and in the process assign to each molecule a unique barcode. In turn, this barcode is comprised of the unique combination of two of multiple different fluorescent dyes, which can overlap in color or fluorescent lifetime but not both. By leveraging color- and lifetime-resolved measurements, the authors can then distinguish the specific dyes associated with each RNA and, thus, identify the RNA molecule.

The authors validate their approach by measuring 10 genes in cell culture, where they show relatively low rates of false measurements in companion controls and strong correlation between the RNA abundances determined via MOSAICA and via bulk RNA-sequencing. They also validate MOSAICA by measuring 6 genes in formalin-fixed paraffin-embedded (FFPE) melanoma samples. Again, they show relatively low rates of false detection and an overall agreement between abundances determined via MOSAICA and bulk RNA-sequencing.

Image-based approaches to spatial transcriptomics represent an exciting and growing set of tools, and, in this respect, the introduction of time-resolved imaging into this toolbox represents a novel and important contribution. For this reason, we are supportive of the publication of this manuscript; however, there are a few issues that we feel should be addressed in revision to both fix a few minor errors and better benchmark the performance of this technique.

Answer: We'd like to thank this Reviewer for his/her overall positive feedback.

First, while the authors have provided a well described set of proof-of-principle and benchmarking experiments, there are several aspects of MOSAICA that are not described, measured, or quantified that are likely to be of interest to anyone considering adopting their technique.

1) Could the authors provide measures of reproducibility in their measurements? For example, how well do the abundance values determined via MOSAICA in replicate measurement in cell culture or in the melanoma sample correlate with one another? At least two replicate measurements for each experiment should be presented for a new technique.

Answer: Following the Reviewer’s suggestion, we have run 2 additional experimental replicates for the cell culture experiments (Figure 4) as well as 1 additional experimental replicate for the melanoma FFPE tissue sample (Figure 5), each accompanied with 2 negative controls. We currently have a total of 5 fields of view for each multiplexed experiment, giving a factor of five in the total puncta count and cell statistics. We have updated Figures 4 and 5 accordingly with supplemental data; correlation plots now present error bars accounting for the replicates. We have also created a new Supplementary Figure 5 with the individual results of the replicates. In summary, the replicates for both the cell culture ($r = 0.96$ between puncta count vs sequencing; $r = 0.94$ among replicates) and tissue experiments ($r = 0.97$ between puncta count vs sequencing; $r = 0.96$ among replicates) were highly consistent with the previous results.

Modified sections of Figures 4 and 5:

Figure 4. **h**) Number of puncta detected for each target gene expression in each cell for the labeled 10-plex samples. **i**) Mean puncta counts per cell for each gene in the 10-plex samples (left, $n=3$ experimental replicates, 364 total cells profiled) and negative control probe samples (right, $n=3$ experimental replicates, 189 total cells profiled). **j**) Correlation of detected puncta (mRNA puncta count) vs. RNA-bulk sequencing (normalized counts) is shown for each target yielding a correlation (Pearson r) of 0.96.

Figure 5. **g**) Mean puncta counts per cell of each gene in the 6-plex sample ($n=2$ experimental replicates, 174 cells). **h**) Puncta count for the negative control probe sample ($n=2$ experimental replicates, 375 cells). **i**) Correlation of detected puncta (mRNA puncta count) vs. bulk sequencing (fragments per kilobase per million) is shown for each target.

We have also modified our methods section to include a “Experimental Replicates” sub-section:

“For the 10-plex cell culture experiments, we ran 3 experimental replicates from which we imaged 6 fields of view (100x100x5 μ m each), with 364 cells in total. In these image stacks, a total of 65,562 puncta were detected where 38,056 were assigned and 27,506 were unassigned to a target. The unassigned counts were further categorized based on assumed overlapping errors (2,439) or as undetermined counts (25,053). In the associated negative controls, a total of 3 experiments were performed, of which we imaged 4 fields of view containing 189 cells total. In these experiments, a total of 2,034 puncta were detected, of which 61 were classified as targeted genes due to the expected spectral and lifetime signature and 1,959 were classified as undetermined.

The tissue experiments with a 6-plex gene panel were replicated a total of 2 times yielding 2 fields of view of 130x130x3 μ m each, together containing 174 cells. A total of 2,934 puncta were detected of which 1,770 were assigned to a target and 1,164 were unassigned, the latter group divided into 62 puncta unassigned due to overlap and 1,100 labelled as undetermined. In the associated negative controls, we ran a total of 3 experiments yielding 3 fields of view and 375 cells. In these fields of view, 390 puncta were detected, of which 43 were assigned to targeted genes. Of the other 347, only 4 were assigned to overlap and 339 to undetermined.

Additional 8-plex and 2-plex experiments were performed on cell cultures, two replicates each, yielding a total of 143 and 130 profiled cells, respectively. Quantification of the experimental replicates by means of cross correlation is presented in Supplementary Figs. 5 and 6.”

2) The authors provide a nice benchmark measurement for the rate of false positives in which they leverage FISH probes that do not target RNAs in their samples. However, the false positive rate measured in samples in which there are no bright FISH signals may not be representative of the real false positive rates in a full MOSAICA measurement. For example, such measurements may not capture the rate at which one bright RNA is misidentified as another bright RNA. Could the authors provide a measure of their false positive rate that captures this aspect? A straightforward measurement of this property should be possible with their existing reagents. For example, they could repeat the cell culture experiments but leave out the probes to one or two genes. The rate at which these genes are detected would then provide this additional false positive rate. Such a measurement would have the added benefit of more closely reflecting the standard approach for estimating false positives in other image-based transcriptomic techniques, i.e., MERFISH, seqFISH, starMAP etc.

Answer: We thank the Reviewer for his/her idea to better assess the false positive rate of our assay. Following this approach, we performed 2 additional experiments with an 8-plex as well as 2 additional experiments with a 2-plex panel to compare the detected transcript abundance values and correlation coefficients with the previous 10-plex experiment. We have added an

additional Supplementary Fig. 6 to the manuscript which presents these data. The new results from 8-plex and 2-plex are in very much accordance with the 10x, suggesting that our previous determination of false positive rate was correct and potential misidentification of bright puncta was not a major contributor to false positives. Specifically, we compared the mean detected puncta counts for each gene obtained in the 10-plex, 8-plex, and 2-plex experiments to RNA sequencing data (Supplementary Figure 6a). We observed a Pearson correlation of 0.96 and 0.97, for the 10-plex and 8-plex experiments, respectively, to RNA sequencing data. A Pearson correlation was not performed for the 2-plex experiments because there were only 2 points (genes). Instead, a binomial test between the 2-plex and 10-plex experiments were performed, $p=0.0796$, showing close agreement. We also compared the puncta counts per cell of each of the 8 genes to those obtained in the replicates for the 10x panel, obtaining a Pearson correlation of 0.98 (Supplementary Figure 6b). The comparison of the average counts per cell for each gene was also done by plotting mean detected puncta values and error bars with standard deviation to reveal the new experiments falling within the error bars of the original 10x panel (Supplementary Figure 6c,d).

Supplementary Fig. 6. Assessing detection rates of genes in the 10-plex experiments with 8-plex and 2-plex experiments. a) Mean puncta counts per cell for each gene in the 10-plex, 8-plex and 2-plex experiments against RNA sequencing data. A Pearson correlation of 0.97 and 0.96 for the 10-plex and 8-plex to RNA sequencing data are shown, respectively. b) Mean puncta counts of the 8-plex experiments were correlated with the 10-plex experiments, Pearson $r = 0.98$. c) Plotted mean puncta counts per cells for each gene obtained from the 10-plex, 8-plex and 2-plex experiments for comparison. d) Detail of the two genes tagged in all three sets of experiments. Binomial test between the counts of the 2-plex and expected proportion from the 10-plex gives $p=0.0796$.

We have modified paragraph 4 under “Simultaneous 10-plex mRNA detection in fixed colorectal cancer SW480 cells using MOSAICA” to include:

“Furthermore, to assess the rate of false positives and determine if one bright mRNA target could potentially be misidentified as another target, we repeated our experiment by leaving out probes for some genes and then comparing the detection rate of remaining targets with the 10-plex data. Specifically, we performed 2 additional experiments with an 8-plex as well as 2 additional experiments with a 2-plex panel to compare the detected transcript abundance values and correlation coefficients against

the 10 plex sample (Supplementary Fig. 6). We observed that there were no significant differences between these panel sizes in terms of target detection rate, indicating that target misidentification was not an issue for these panel sizes.”

3) The authors contrast MOSAICA to other image-based techniques and highlight the potential for MOSAICA to be faster. However, these other techniques, MERFISH/seqFISH/starMAP, are compatible with wide-field imaging whereas MOSAICA is likely constrained to point scanning. The authors should provide some information on how long it takes to collect an image of a given size so that readers can better judge the potential throughput of this technique.

Answer: We agree with the Reviewer that wide-field imaging is considerably faster than scanning-based techniques. However, we would like to point out that, although we used a scanning confocal microscope for the main results in this paper, our technique is not limited to scanning-based instruments. While our technique does require sufficient image pixel sizes, axial resolution, and photon counts to work, this can be accomplished with a hyperspectral light sheet camera-based system which we have previously successfully implemented for a different application, or with a spinning-disc confocal instrument equipped with a FLIM camera to greatly increase imaging throughput. We have modified paragraph 7 in the Discussion section to provide information of our typical imaging speed and make it clearer that MOSAICA is not restricted to scan-based instruments, and provided some examples of wide-field imaging techniques which should be compatible with certain modifications:

“Moreover, the current scanning confocal microscope implemented in MOSAICA can achieve high spatial resolution and z-sectioning but is limited by a relatively longer imaging time. As an example, each z-slice of our spectral-FLIM images (figures 4 and 5) took around 1.5min (1024x1024 images at 16 μ s pixel dwell time, accumulating an average of 6 frames). However, we anticipate that this approach is compatible with any wide-field imaging technique as long as sufficient pixel number, axial resolution, and photon counts are met. This can be accomplished with our recently developed camera-based light sheet imaging system⁵¹ or a spinning disc confocal system equipped with a FLIM camera to greatly improve imaging throughput⁵².”

51. Hedde, P.N., Cinco, R., Malacrida, L. et al. Phasor-based hyperspectral snapshot microscopy allows fast imaging of live, three-dimensional tissues for biomedical applications. *Commun Biol* 4, 721 (2021).
52. Buranachai, C., Kamiyama, D., Chiba, A., Williams, B. D. & Clegg, R. M. Rapid frequency-domain flim spinning disk confocal microscope: Lifetime resolution, image improvement and wavelet analysis. *J. Fluoresc.* **18**, 929–942 (2008).

4) It may be worthwhile for the authors to note that their high correlation coefficient between bulk RNA-seq and MOSAICA for their melanoma measurements would appear to be dominated by the measurement of one, highly abundant gene: POLR2A. The correlation within the other genes is likely far more modest. FFPE can be challenging sample source for FISH measurements, so this issue is likely not reflective of a problem with MOSAICA. Nonetheless, the authors may wish to better explain this high correlation coefficient.

Answer: The Reviewer raised an excellent point. While the correlation in the cell culture sample (Figure 4J) between bulk RNA-seq and MOSAICA shows close agreement for many markers, the correlation in the FFPE tissue sample (Figure 5I) is particularly dependent on the presence of POLR2A and becomes weaker though still positive ($r = 0.44$) for the other markers if we exclude POLR2A. We agree with the Reviewer that one major factor which can contribute to this disparity is the different preservation and pretreatment methods used to process these samples. In particular, there have been multiple studies documenting the effects of FFPE processing on mRNA degradation and fragmentation which can compromise the results of absolute mRNA quantification in FFPE tissues⁴⁴⁻⁴⁶. Some workarounds include normalizing overall counts to reference tissue samples with a highly expressed gene such as POLR2A but the issue with lowly expressed genes being lost or underrepresented during this processing step remains a critical challenge in the field. Furthermore, storage time, storage temperature, and handling conditions for these tissue blocks is a highly critical parameter for downstream performance of RNA-FISH assays which may at times be difficult to account for by the user. We have acknowledged this limitation and incorporated this discussion in revised manuscript:

“We acknowledge that this strong correlation is particularly dependent on the presence of the highly abundant POLR2A gene. The correlation for the other lower expressed targets excluding POLR2A is $r=0.44$ which, although still positive, is weaker. We attribute this discrepancy to preanalytical variables typically associated with FFPE sample preservation and pretreatment. For instance, there have been multiple studies which documented increased variability in quantifying lowly expressed genes in FFPE tissues due to RNA degradation or cross-linking of proteins with nucleic acids⁴⁴⁻⁴⁶.”

44. Klopffleisch, R., Weiss, A. T. A. & Gruber, A. D. Excavation of a buried treasure - DNA, mRNA, miRNA and protein analysis in formalin fixed, paraffin embedded tissues. *Histol. Histopathol.* 26, 797–810 (2011).
45. Ripoli, F. L. et al. A comparison of fresh frozen vs. Formalin-fixed, paraffin-embedded specimens of canine mammary tumors via branched-DNA assay. *Int. J. Mol. Sci.* 17, (2016).
46. Scicchitano, M. S. et al. Preliminary comparison of quantity, quality, and microarray performance of RNA extracted from formalin-fixed, paraffin-embedded, and unfixed frozen tissue samples. *J. Histochem. Cytochem.* 54, 1229–1237 (2006).

5) Finally, for their next step, the authors aim to expand MOSAICA to a 60-plex scheme that can detect 1770 mRNA species at once. However, the multitude of fluorophores will likely cause the spectral and lifetime phasor spaces to become overcrowded, thus increasing measurement error rates. If the authors would like to propose that 60-plex (or higher multiplexing in general) is possible, it would be helpful to provide some rationale as to what number of distinct fluorophores could be observed and the practical and technical limitations on this number.

Answer: We agree with the Reviewer that one challenge in scaling up multiplexing in our case is the crowding of distributions in the phasor space. Nevertheless, we believe it is both theoretically and practically feasible for us to achieve up to approximately 60-plex or even higher. We would like to emphasize that in our aim, 60 is the number of combinations, not the number of fluorophores. In the Discussion, we have clarified and acknowledged this issue starting with paragraph 4 and have provided a theoretical framework as to how we are able to scale up multiplexing:

“One common challenge in imaging-based spatialomics analysis is optical crowding which can limit both the number of molecules that can be detected and the detection efficiency and accuracy. For instance, as we scale up multiplexing capabilities by labeling more mRNA and proteins with additional fluorophores, more labeled targets and their fluorescent combinations will begin occupying the same voxel, leading to challenges in determining both how many targets there are as well as which type of targets are present within each voxel. We have modeled this phenomenon in Equation 2 (Methods section) and plotted the results in Supplementary Figure 4. Based on our estimates and in our current transcript density conditions, overlapping accounts for only around 6% of the detected puncta. We currently do not further resolve these cases and, instead, simply categorize them into the overlapping group, which do not contribute to total counts. In addition, we intend to further address these cases in the future using our novel multicomponent approach³⁶ to unmix spectral/lifetime components within a single voxel by means of higher harmonics of the phasor transform.

With respect to the crowding issue, the phasor analysis method has an additional limitation related to the use of the combinatorial technique. Even if the isolated fluorescent dyes are very far apart on the phasor space, their combinations fall inside the polygon as determined by the positions of the individual dyes. As one increases the number of dyes, the combinations start to overlap creating an ambiguity. For this reason, as we increase our multiplexed panel, our strategy is not only to employ labels which are distinctly separated by both spectral and lifetime properties but importantly to also use more combinations of different labels rather than using more combinations of the same labels. The tradeoff between these two counteracting parameters is an exciting endeavor which we look forward to exploring as we progressively build up our repertoire of fluorophores.

Regarding the phasor population overlap, given the imaging settings we have used for the experiments in this paper, the signal-to-noise ratio produces gaussian phasor distributions with 99.7% of the pixels within 0.01 phasor units (6σ). Although the distributions tend to converge due to background autofluorescence, their small covariance matrices guarantee a high level of confidence in assigning each pixel to the correct cluster. As an example, the three gaussian distributions in Fig. 3h, have covariance matrices of (coordinate S first, G second) $\Sigma_1 = \begin{pmatrix} 12 & -5 \\ -5 & 89 \end{pmatrix} \times 10^{-5}$, $\Sigma_2 = \begin{pmatrix} 6 & -8 \\ -8 & 51 \end{pmatrix} \times 10^{-5}$ and $\Sigma_3 = \begin{pmatrix} 12 & -8 \\ -8 & 50 \end{pmatrix} \times 10^{-5}$ and mean coordinates of $\mu_1 = \begin{pmatrix} 0.26 \\ 0.40 \end{pmatrix}$, $\mu_2 = \begin{pmatrix} 0.25 \\ 0.49 \end{pmatrix}$ and $\mu_3 = \begin{pmatrix} 0.23 \\ 0.57 \end{pmatrix}$. With these values, the distance between the leftmost and right most cluster is of 0.17 phasor units, with the mean standard deviation from the covariance matrices being 400 times smaller at 3.65×10^{-4} . With numbers like these, we anticipate that our clustering technique can easily resolve even more challenging scenarios such as 6 lifetime phasor clusters and 10 spectral phasor clusters. As a result, our next immediate goal is to scale our multiplexing capability by detecting around 60 mRNA targets simultaneously with 12 different fluorophore species within the same sample. We aim to use 7 spectrally distinct fluorophores and an additional 5 with overlapping spectra but are resolvable by lifetime. A combinatorial scheme of 12 choose 2 would yield 66 combinations. We could resolve these combinations using a 7-spectral channel instrument where 5 of the channels would present 3 populations in the lifetime phasor plot (one for each of the two probes with overlapping spectra in that channel plus the third being the combination of the two). Looking another step ahead, by implementing our recently developed 32-channel spectral-FLIM detector³⁴ which can provide 32 independent spectral sources with 6 lifetime clusters per channel, 192 different fluorophore species can be accessed to provide significantly higher plex detection capabilities.”

34. Scipioni, L., Rossetta, A., Tedeschi, G. & Gratton, E. Phasor S-FLIM: a new paradigm for fast and robust spectral fluorescence lifetime imaging. *Nat. Methods* **18**, 542–550 (2021).
36. Vallmitjana, A. *et al.* Resolution of 4 components in the same pixel in FLIM images using the phasor approach. *Methods Appl. Fluoresc.* **8**, 035001 (2020).

Second, while we understand the desire to set MOSAICA in the most favorable light relative to other techniques, we have several concerns with the way in which prior work is described. There is a growing array of techniques that fall into the category of spatially resolved transcriptomics, yet the description and grouping of these techniques is not always accurate, and fundamental techniques are not mentioned.

1) The authors group together single-molecule imaging techniques, e.g., MERFISH, seqFISH, RNAscope, and SABER, with spatial capture methods, e.g., GeoMx, slide-seq. These technologies are not easily compared both in resolution, performance, and technical requirements. Perhaps it would be clearer to simply focus the comparison on imaging-based techniques, which is the category in which MOSAICA clearly falls.

Answer: Per Reviewer's suggestion, we have separately discussed (a) imaging- and FISH-based spatial transcriptomics, and (b) sequencing-based spatial transcriptomics in the Introduction. We have also cited a comprehensive review on spatial transcriptomics that came out very recently (Lewis, S.M., et al., Spatial omics and multiplexed imaging to explore cancer biology. Nature Methods, 2021 Sep;18(9):997-1012. Ref. 22) for interested readers who want to learn more depth of each of these technologies. We have modified our introduction section (paragraph 2) to reflect this change:

“Imaging- and FISH-based spatial transcriptomic methods that employ sequential labeling, stripping, and imaging (e.g., seqFISH, MERFISH) or branched amplification (e.g., RNAscopeTM, SABER) are often complicated, error-prone, time-consuming, laborious and/or costly to scale up^{18–22}. Furthermore, repeated processing of the same sample can in some cases damage tissue structural integrity and target molecules and may not always be amenable for clinical applications such as profiling patient biopsies. Spatial transcriptomics using in situ sequencing (e.g., ISS, FISSEQ, starMAP and ExSeq) or in situ barcoding coupled with ex situ sequencing (e.g., GeoMx, slide-seq, and DBiT-seq) can drastically improve multiplexing but suffer from reduced spatial resolution and detection efficiency especially for low-abundance targets^{22–25}.”

Furthermore, we have edited the introduction (paragraph 3) to emphasize MOSAICA as an imaging-based technique:

“In this work, we report a new **fluorescence imaging-based** spatial-omics technology....”

2) Moreover, the authors call GeoMx and slide-seq in-situ sequencing techniques, yet this definition is not the standard categorization in the field. These techniques do not sequence molecules in situ but rather incorporate spatial barcodes with cDNA created from captured molecules which are then removed and sequenced ex situ to reconstruct molecular identity and location. The more standardly defined in situ sequencing techniques by Nilsson (ISS), Church (FISSEQ), Deisseroth (starMAP), and Boyden (ExSeq) are not mentioned, and if the authors feel it is important to contrast MOSAICA with other approaches for multiplexed single-RNA-molecule identification, they should probably mention these methods.

Answer: Addressed. Please see above.

3) It is not accurate to describe all multiplexed FISH techniques as involving stripping or causing damage to tissues. Some techniques, such as MERFISH and a few newer implementations of seqFISH, do not remove secondary (readout) probes from primary (encoding) probes but rather chemically cleave or inactivate fluorophores. The former approach is proven to be quite gentle, and samples are stable through tens of rounds of this process, in terms of both tissue structure and mRNA quality. A major technical advantage of MOSAICA is the fact that repetitive sample processing is not needed, but it is important to note that methods exist for this repetitive processing to maintain sample integrity. In short, sample degradation is not a drawback of all repetitive processing methods.

Answer: We have corrected our statement regarding this claim by revising the introduction section (end of paragraph 2):

“Furthermore, repeated processing of the same sample can in some cases affect tissue structural integrity and target molecules and may not always be amenable for clinical applications such as profiling patient biopsies.”

We have also modified our Discussion section (paragraph 2) to:

“This can be particularly important in clinical settings, where biopsy samples are limited in quantity and the capability to preclude repetitive sample processing can simplify overall clinical workflow.”

4) There are several places in the manuscript where concrete numbers are provided for potential improvement of MOSAICA over other techniques (i.e., “For a 10-plex panel, this drastically shortens assay time by at least of factor of 4-5x and scales up even more as the panel becomes more multiplexed...”) In situations such as this one, these arguments should be supported with estimates drawn from the literature for the performance of existing techniques and measurements of the performance of MOSAICA. For example, as a point-scanning method it is possible that image acquisition is much slower than wide-field methods, so the enhancement of speed that comes with the removal of repetitive staining is not obvious.

Answer: After much deliberation, we decided to use descriptive rather than quantitative comparisons as the latter would indeed require head-to-head experiments conducted under the same settings (e.g., same sample preparation, same targets). We have modified our Discussion accordingly:

“Compared to existing sequential hybridization and imaging approaches, MOSAICA significantly reduces the number of hybridization and imaging rounds required to profile larger multiplexed panels of RNA biomarkers. This can be particularly important in clinical settings, where biopsy samples are limited in quantity and the capability to preclude repetitive sample processing can simplify overall clinical workflow. In terms of

cost, MOSAICA utilizes inexpensive DNA primary probes which can be purchased in batch or as microarrays for a minimal price. Fluorescently conjugated secondary probes can also be used and shared as a common set among many different genes, scaling down costs to several dollars per assay.”

5) In several locations it is stated that MOSAICA is an error-correcting technique, yet that statement does not appear to be true. Error correction formally requires that a measurement that is incorrect can be associated with a correct barcode. Removal of noise sources or background are not considered error correction. The encoding scheme utilized in this work is a n-choose-k scheme in which k is 2. This scheme can detect errors, i.e., any spot that only has one measured dye instead of two clearly had a measurement error, but it does not have a way to predict the fluorophore that was lost in this case. In addition, if an mRNA spot had two fluorophores and one of them is incorrectly detected as a wrong dye, this coding scheme will not detect any error but simply decode the mRNA as an incorrect species. Thus, these codes have limited error-detecting capacity but not error correcting capacity.

Answer: We agree with the Reviewer that the more appropriate term for our approach would be error-detection. As biophysicists, we are accustomed to using the term error-correction for methods for measurement and analysis such as calculating fluorescent lifetimes (e.g. time correlated single photon counting, phasor approach) and background subtraction (e.g. removal of instrument noise and autofluorescence). However, in the context of spatial transcriptomics (e.g. barcoded labeling schemes), error-detection would be more suitable for our approach. We have revised the following sections to reflect this:

(Abstract):

“By integrating the time dimension with conventional spectrum-based measurements, MOSAICA enables direct and highly-multiplexed in situ spatial biomarker profiling in a single round of staining and imaging while providing **error detection** and removal of background autofluorescence... We then showcase MOSAICA’s multiplexing scalability in detecting 10-plex targets in fixed colorectal cancer cells using combinatorial labeling of only five fluorophores with facile **error-detection** and removal of autofluorescent moieties.”

(Introduction, paragraph 4):

“We further demonstrated MOSAICA’s utility in improved multiplexing, **error-detection**, and autofluorescence removal in highly scattering and autofluorescent clinical melanoma FFPE tissues, demonstrating its potential use in tissue for cancer diagnosis and prognosis.”

(Results section under “Simultaneous 10-plex mRNA detection in fixed colorectal cancer SW480 cells using MOSAICA”, 1st and 3rd paragraphs):

“Here, we selected this model as a validation platform to demonstrate the multiplexing scalability and **error detection** capabilities of our approach.”

“MOSAICA employs an **error-detection** based strategy that gates for specific and pre-encoded fluorophore combinations and rejects any fluorescent signature that does not meet these criteria.”

(Results section under “Multiplexed mRNA analysis in clinical melanoma skin FFPE tissues”, 1st paragraph):

We next investigated whether MOSAICA can provide multiplexed mRNA detection with phasor-based **background correction and error-detection capabilities** to clinically relevant and challenging sample matrices.

(Discussion, 1st paragraph):

MOASICA accomplishes this by uniquely integrating the lifetime dimension with the conventional spectral dimension, employing combinatorial fluorescence spectral and lifetime target encoding, and exploiting machine learning- and phasor-based deconvolution algorithms to enable high-plex analysis with **error-detection**.

Reviewer #2

In the manuscript “Spatial transcriptomics using combinatorial fluorescence spectral and lifetime encoding, imaging and analysis” from Vu, Vallmitjana, Gu et al, the authors describe a new method combining in situ labeling of mRNA and protein markers with spectral and lifetime imaging and machine learning.

The novelty of the work is the combination of labels with different lifetimes/spectra with dual-domain spectra-FLIM imaging (application of the s-FLIM technique from the Gratton group by Scipioni et al (Nature Methods)) and machine learning phasor clustering to spatially multiplex transcriptomics data.

There are limited solutions for performing parallel multi-label spatial transcriptomics, mostly owing to the challenges imposed by the labels’ spectral emission overlap. Here the authors exploit imaging of fluorescent spectral and lifetime domain to provide high-throughput information-rich imaging data.

The authors claim that the MOSAICA approach:

1. Can spatially reveal and visualize the presence, identity, expression level, location, distribution and heterogeneity of each target mRNA in the 3D context
2. Can perform 10-plex imaging with error -correction and autofluorescence removal

The first claim is supported with examples and analysis of experimental data. The second claim is supported by comparison of real data with “artificially filtered” data, simulations and comparison with standard analysis approaches.

In all cases, the authors showcase the capabilities of MOSAICA in multiplexing, providing some quantitative parameters. According to the experimental data the performance of MOSAICA is promising, with 4-5x assay time shortening and 10-20x cost reduction.

In summary, the paper presents a novel technique and, if the claims stated by the authors are confirmed (see my major comments below), it represents an original and significant technical advance for spatial transcriptomics. I think it is of interest and of potential impact for several applications, especially for diagnostics.

For these reasons I think that the paper is of great interest for Nature Communications and well-worth of publication on this journal.

Answer: We'd like to thank this Reviewer for his/her overall positive feedback.

However, there are some fundamental points that must be addressed to understand the true advantages of the proposed method. The quantification of performance of the technique is partially present in the manuscript, although scattered between main text and supplementary. The performance/repeatability of the combinatorial labels in the aspect of instrumental noises is lightly described and not well highlighted for a broad viewership.

Finally, the manuscript writing is acceptable and, with minimal improvements, can cater to the broad audience of Nature Communications.

I ask the authors to address the following concerns.

Major comments

1. The authors lightly describe the performance of the combinatorial labeling in the aspects of label-implementation, detection and noises of the system. The Supplementary Figure 4 reports an informative description, and one of the few quantifications, of the performance of the MOSAICA (Supp. Figure 4 C,D). In this figure 56% of puncta are estimated to be assigned correctly, 37% is undetermined and 6% is overlapping. The main text only references the 6% lost due to overlap. The 37% undetermined should be reported more clearly and discussed in the main text as most readers will be interested in this information.

Answer: We agree with the Reviewer that this is an important topic which was not sufficiently discussed in the original manuscript. We have modified the following passage in our results section (under Simultaneous 10-plex mRNA detection in fixed colorectal cancer SW480 cells using MOSAICA, 3rd paragraph) to expand upon this topic including potential sources contributing to this group:

“MOSAICA employs an error-detection strategy that gates for specific and pre-encoded fluorophore combinations and rejects any fluorescent signature which do not meet these criteria. For instance, of the total detected puncta ($n = 65,562$), we observed a considerable fraction of puncta, $n = 25,053$ (38%), which was rejected based on their fluorescence emission of only a single channel (Supplementary Fig. 4C). We characterize

this group as the “undetermined group” because each event can belong to: 1) the nonspecific binding of probes, 2) autofluorescent moieties, or 3) mRNA transcripts which were not fully labeled with both dyes. For the first case, as previously characterized by several groups, nonspecific binding events is a common inherent issue with single-molecule FISH techniques which arises from the stochastic binding of DNA probes towards cellular components such as proteins, lipids, or nonspecific regions of RNA and follow a random distribution^{14,20}. When combined with events which may be autofluorescence moieties (e.g. porphyrins, flavins) which can exist as isolated diffraction-limited structures and emit strong fluorescence in any particular single channel³⁸ or mRNA transcripts which were labeled with only one set of fluorophores, these groups represent a confounding issue for standard intensity-based measurements and analysis because they share similar SNR and intensities to real labeled puncta and cannot be differentiated without additional lengthy or complex techniques such as sample clearing or iterative-based labeling and imaging error-correction³⁹. Therefore, the main benefit of implementing the combinatorial encoded criteria is to ensure target detection fidelity by rejecting stochastic and nonspecific binding labeling events as well as any event eliciting a lifetime signature that deviated from the utilized fluorophores.”

14. Moffitt, J. R. et al. High-performance multiplexed fluorescence in situ hybridization in culture and tissue with matrix imprinting and clearing. *Proc. Natl. Acad. Sci. U. S. A.* **113**, 14456–14461 (2016).
20. Eng, C. H. L. et al. Transcriptome-scale super-resolved imaging in tissues by RNA seqFISH+. *Nature* **568**, 235–239 (2019).
39. Wang, G., Moffitt, J. R. & Zhuang, X. Multiplexed imaging of high-density libraries of RNAs with MERFISH and expansion microscopy. *Sci. Rep.* **8**, 1–13 (2018).

2. In this work, labels with different lifetime/spectra are combined ad hoc to fill a discretized matrix in the lifetime/spectra domains (as shown in Figure 1 F). The repeatability of this combinatorial label will determine the precision of the multiplexing. Example: the combination that falls in Target6 (Figure 1F) needs to be sufficiently different from the combination that falls in Target 7. Did the authors characterize the uniformity of a single combinatorial label? Example: measuring only the combination of lifetime/spectra that provides Target6 (Figure 1F), what is the average lifetime/spectra and their respective standard deviation? The broadness of this standard deviation will affect both accuracy of measurement and the total multiplexing capability. The authors should discuss this in the main text.

Answer: We agree with the Reviewer that this is an important topic which needs further clarification in the manuscript. We have characterized each label both independently and as combinations with other fluorophores. We did this in dye solutions to determine which labels

had distinct locations on the phasor space and in 1- and 2-plex cell culture experiments to verify that these independent populations could be resolved *in situ*. Furthermore, the broadness of the gaussian distributions on the phasor space are dependent on the ratio between the baseline instrumental/biological noise and the number of photons collected. This in turn depends on the particular fluorescent probe and the imaging settings. We determined these settings empirically in order to have a relatively low imaging acquisition time while collecting enough photons to minimize the variance of the phasor distributions and allowing them to be distinguished with our phasor clustering technique. Once the instrument and imaging settings were kept identical across all measurements, the phasor distributions remained consistent. Lastly, we would like to reiterate that in the experiments shown in the paper, there were up to three lifetime populations in each lifetime phasor plot and that our phasor clustering method allows for a fourth cluster that accounts for background noise. We have added a paragraph 6 to the discussion section of the main text to clarify this point by providing the exact numbers on the covariance matrices of the phasor distributions:

“Regarding the phasor population overlap, given the imaging settings we have used for the experiments in this paper, the signal-to-noise ratio produces gaussian phasor distributions with 99.7% of the pixels within 0.01 phasor units (6σ). Although the distributions tend to converge due to background autofluorescence, their small covariance matrices guarantee a high level of confidence in assigning each pixel to the correct cluster. As an example, the three gaussian distributions in Fig. 3h, have covariance matrices of (coordinate *S* first, *G* second) $\Sigma_1 = \begin{pmatrix} 12 & -5 \\ -5 & 89 \end{pmatrix} \times 10^{-5}$, $\Sigma_2 = \begin{pmatrix} 6 & -8 \\ -8 & 51 \end{pmatrix} \times 10^{-5}$ and $\Sigma_3 = \begin{pmatrix} 12 & -8 \\ -8 & 50 \end{pmatrix} \times 10^{-5}$ and mean coordinates of $\mu_1 = \begin{pmatrix} 0.26 \\ 0.40 \end{pmatrix}$, $\mu_2 = \begin{pmatrix} 0.25 \\ 0.49 \end{pmatrix}$ and $\mu_3 = \begin{pmatrix} 0.23 \\ 0.57 \end{pmatrix}$. With these values, the distance between the leftmost and right most cluster is of 0.17 phasor units, with the mean standard deviation from the covariance matrices being 400 times smaller at 3.65×10^{-4} . With numbers like these, we anticipate that our clustering technique can easily resolve even more challenging scenarios such as 6 lifetime phasor clusters and 10 spectral phasor clusters. As a result, our next immediate goal is to scale our multiplexing capability by detecting around 60 mRNA targets simultaneously with 12 different fluorophore species within the same sample. We aim to use 7 spectrally distinct fluorophores and an additional 5 with overlapping spectra but are resolvable by lifetime. A combinatorial scheme of 12 choose 2 would yield 66 combinations. We could resolve these combinations using a 7-spectral channel instrument where 5 of the channels would present 3 populations in the lifetime phasor plot (one for each of the two probes with overlapping spectra in that channel plus the third being the combination of the two). Looking another step ahead, by implementing a 32-channel spectral-FLIM detector³⁴ which can provide 32 independent spectral sources with 6 lifetime clusters per channel, 192 different fluorophore species can be accessed to provide significantly higher plex detection capabilities.”

34. Scipioni, L., Rossetta, A., Tedeschi, G. & Gratton, E. Phasor S-FLIM: a new paradigm for fast

and robust spectral fluorescence lifetime imaging. *Nat. Methods* **18**, 542–550 (2021).

3. The effects of point 2 above are amplified by the presence of noise in the instrument. Measurement of 1 spectrum (or lifetime) on phasor produces a gaussian distributed cluster. Provided n combination of labels, the tails of these clusters will increasingly overlap with the number of combinatorial labels. The authors pick labels that are well separated on phasor to minimize this overlap, however supp. Figure 4 does report some information loss. Based on these negatively affecting components, what is the maximum number of plex that MOSAICA could reliably identify? The authors should discuss this in the main text.

Answer: This again is a great point and we have made an effort to address it in combination with the previous point and provide a theoretical framework to justify the number of plex we expect to attain. Indeed, there is a technical limitation on the number of different fluorophore species which can be utilized. This limitation is dependent on the degree of overlap between the different species on the phasor plot, the number of spectral channels which can be reliably accessed on the visible spectrum and the infrared region, and importantly, the number of available fluorophores that have unique subspace in both spectral and lifetime space. The more labels which we can find that fulfill these criteria, the higher the number of combinations which we can utilize with minimal measurement and analysis error. We attribute this information loss in Supplementary Figure 4 to overlap issues rather than phasor analysis which is addressed in Reviewer 2 question 4. In the revised main text, we have discussed the “undetermined group” and “overlapping group” presented in Supplementary Figure 4. In the Discussion section, when quantifying the gaussian covariance matrices, we have added a justification for our estimated 60 plex achievement and a more ambitious 200 plex (shown above in response to #2).

4. How do the authors account for when two combinatorial labels fall within the same pixel? E.g. looking at figure 3 H, if one magenta label and one green label fall within the same pixel the apparent signal will be orange, providing an incorrect assignment of the label. This problem increases with the number of possible target genes (equation 1), quadratically. The authors should discuss this in the main text.

Answer: We agree with the Reviewer that one common challenge in imaging-based spatialomics analysis is optical crowding which can limit both the number of target molecules that can be detected and the detection efficiency/accuracy of our assay. Currently, we do not directly resolve the case of several transcripts in the same voxel. Instead, we detect and categorize these cases into a different group (overlapped). In the example given (see embedded

figure), if only 1 transcript occupy a particular voxel and is labeled with green and magenta labels, its phasor space will converge only at the location of the orange cluster but not at the locations of the magenta or green clusters, making it easy to discriminate, e.g., MKI67. However, if an additional transcript with a green and a different second label such as a red label from a different spectra channel is also present, it will

be difficult to assign which two transcripts were actually present, e.g. green-magenta, red-magenta, vs. green-red. In future work, we plan on implementing our recently developed multicomponent analysis approach³⁶ to unmix spectral/lifetime components within a single voxel by means of higher harmonics of the phasor transform to address this issue.

We have added two paragraphs in the Discussion section to acknowledge these issues and discussed some strategies to pursue:

“One common challenge in imaging-based spatialomics analysis is optical crowding which can limit both the number of molecules that can be detected and the detection efficiency and accuracy. For instance, as we scale up multiplexing capabilities by labeling more mRNA and proteins with additional fluorophores, more labeled targets and their fluorescent combinations will begin occupying the same voxel, leading to challenges in determining both how many targets there are as well as which type of targets are present within each voxel. We have modeled this phenomenon in Equation 2 (Methods section) and plotted the results in Supplementary Figure 4. Based on our estimates and in our current transcript density conditions, overlapping accounts for only around 6% of the detected puncta. We currently do not further resolve these cases and, instead, categorize them into the overlapping group, which do not contribute to total counts. In addition, we intend to further address these cases in the future using our novel multicomponent approach³⁶ to unmix spectral/lifetime components within a single voxel by means of higher harmonics of the phasor transform.

With respect to the crowding issue, the phasor analysis method has an additional limitation related to the use of the combinatorial technique. Even if the isolated fluorescent dyes are very far apart on the phasor space, their combinations fall inside the polygon as determined by the positions of the individual dyes. As one increases the number of dyes, the combinations start to overlap creating an ambiguity. For this reason, as we increase our multiplexed panel, our strategy is not only to employ labels which are distinctly separated by both spectral and lifetime properties but importantly to also use more combinations of different labels rather the same labels. The tradeoff between these two counteracting parameters is an exciting endeavor which we look forward to exploring as we progressively build up our repertoire of fluorophores.”

Minor comments:

5. In introduction section, line 122 “we were able to discriminate a large repertoire of spectral and lifetime components”, providing an actual number would highly improve the impact of the work.

Answer: We have modified this sentence to:

“By utilizing both time and spectral domains for labeling and imaging, we were able to discriminate a repertoire of 10 different fluorescent signatures against autofluorescent moieties and nonspecific binding events within the same sample in this study and expect to scale up to at least 60-plex in the future to enable increased multiplexing capabilities with standard optical systems.”

6. Line 151 “imaged using a confocal microscope (Fig. 1D) equipped with spectral and lifetime capabilities”, if this is the same instrument described by Scipioni et al (Nat. Methods 2021), please reference the work, else if this is the SP8-falcon mentioned in Methods section, add “commercial unit” or similar.

Answer: This section was intended to describe the generic MOSAICA workflow and its minimal system requirements. For the bulk of the work presented in this manuscript, we utilized the Leica SP8 Falcon system. We have reworded the sentence to:

“The labeled samples are then imaged using a custom built or commercial microscope (e.g., the Leica SP8 Falcon used in this study) equipped with spectral and lifetime imaging capabilities (Fig. 1D).”

Reviewers' Comments:

Reviewer #1:

Remarks to the Author:

I found the first draft of this manuscript to be a strong contribution, and, with the revisions provided by the authors, I find the current draft to be even stronger. I fully support publication on this excellent manuscript, and I think the authors for such a detailed and thorough response to the concerns I raised.

Reviewer #2:

Remarks to the Author:

The authors have replied in detail to all my major comments and amended to the minor points I suggested.

In this revised version, Vu and coworkers have substantially improved the manuscript. The additions of new material, data and repeats in Main manuscript, as well as additional Supplementary Figures, provide a clearer description of the capabilities and performance of this novel technology. This improved version will help the reader appreciate the novelty and efficacy of MOSAICA method.

The authors also included experimental details and repeats that were missing in the previous version. The quantitative aspect of the analysis has been improved. Authors have made considerable effort in characterizing some technical aspects and limitations, also in response to comments of other referees.

I strongly recommend acceptance by Nature Communications without further revision.

Response:

We thank reviewers' for their time to read the revised manuscript and are delighted to see that they are satisfied with all their comments properly addressed.

REVIEWERS' COMMENTS

Reviewer #1 (Remarks to the Author):

I found the first draft of this manuscript to be a strong contribution, and, with the revisions provided by the authors, I find the current draft to be even stronger. I fully support publication on this excellent manuscript, and I think the authors for such a detailed and thorough response to the concerns I raised.

Reviewer #2 (Remarks to the Author):

The authors have replied in detail to all my major comments and amended to the minor points I suggested.

In this revised version, Vu and coworkers have substantially improved the manuscript. The additions of new material, data and repeats in Main manuscript, as well as additional Supplementary Figures, provide a clearer description of the capabilities and performance of this novel technology. This improved version will help the reader appreciate the novelty and efficacy of MOSAICA method.

The authors also included experimental details and repeats that were missing in the previous version. The quantitative aspect of the analysis has been improved. Authors have made considerable effort in characterizing some technical aspects and limitations, also in response to comments of other referees.

I strongly recommend acceptance by Nature Communications without further revision.